# Pharmacokinetics and Pharmacodynamics of Antibody-Drug Conjugates Administered via Subcutaneous and Intratumoral Routes

**DOI:** 10.3390/pharmaceutics15041132

**Published:** 2023-04-03

**Authors:** Hsuan-Ping Chang, Huyen Khanh Le, Dhaval K. Shah

**Affiliations:** Department of Pharmaceutical Sciences, School of Pharmacy and Pharmaceutical Sciences, The State University of New York at Buffalo, Buffalo, NY 14241, USA

**Keywords:** intratumoral administration, subcutaneous administration, antibody-drug conjugate (ADC), local toxicity, monomethyl auristatin E (MMAE), pharmacokinetics/pharmacodynamics (PK/PD), modeling and simulation (M&S)

## Abstract

We hypothesize that different routes of administration may lead to altered pharmacokinetics/pharmacodynamics (PK/PD) behavior of antibody-drug conjugates (ADCs) and may help to improve their therapeutic index. To evaluate this hypothesis, here we performed PK/PD evaluation for an ADC administered via subcutaneous (SC) and intratumoral (IT) routes. Trastuzumab-vc-MMAE was used as the model ADC, and NCI-N87 tumor-bearing xenografts were used as the animal model. The PK of multiple ADC analytes in plasma and tumors, and the in vivo efficacy of ADC, after IV, SC, and IT administration were evaluated. A semi-mechanistic PK/PD model was developed to characterize all the PK/PD data simultaneously. In addition, local toxicity of SC-administered ADC was investigated in immunocompetent and immunodeficient mice. Intratumoral administration was found to significantly increase tumor exposure and anti-tumor activity of ADC. The PK/PD model suggested that the IT route may provide the same efficacy as the IV route at an increased dosing interval and reduced dose level. SC administration of ADC led to local toxicity and reduced efficacy, suggesting difficulty in switching from IV to SC route for some ADCs. As such, this manuscript provides unprecedented insight into the PK/PD behavior of ADCs after IT and SC administration and paves the way for clinical evaluation of these routes.

## 1. Introduction

Antibody-drug conjugates (ADCs) are becoming increasingly prominent anticancer therapeutics [1]. Currently, 14 ADCs are marketed worldwide for hematological indications or solid tumor cancers [2], of which 11 have gained regulatory approval since 2017 [3]. In addition, over 100 ADCs are under investigation at various stages of clinical development [1,2]. However, the relatively narrow therapeutic index remains one of the main challenges for ADC development. Therefore, strategies that can mitigate off-target toxicities and/or increase the on-target efficacy of these molecules may improve the therapeutic index of ADCs and increase their probability of clinical success [4]. Efforts to enhance the therapeutic index of ADCs primarily focus on optimizing ADC design and construction (i.e., linker chemistry, conjugation method, payload potency, stability) [5,6]. However, approaches that aim to modulate the pharmacokinetics/pharmacodynamics (PK/PD) of ADCs may also have the potential to increase the therapeutic index of ADCs. Given that ADCs comprise large and small molecules, multiple analytes such as the conjugated antibody (mAb), conjugated payload, unconjugated payload, and naked mAb can be measured upon ADC administration [7]. The PK of these analytes in systemic circulation and/or at the site-of-action (i.e., tumor) determine the efficacy and safety of ADCs. Therefore, we hypothesize that switching the routes of administration of ADCs may enable modulation of the PK of individual ADC analytes at the site of efficacy and toxicity, and may provide a viable strategy to modulate the therapeutic index of ADCs.

Currently, all approved ADCs are administered via the intravenous (IV) route, and the therapeutic potential of ADCs via other routes is rarely investigated. However, when it comes to therapeutic proteins such as monoclonal antibodies, while the IV route is still the most commonly used route, several alternative routes of administration have been proven to be successful in improving the therapeutic index of these molecules [8]. Examples of these alternative routes include subcutaneous (SC), intramuscular, intravitreal, intra-CNS routes, inhalation, intra-articular administration, and intratumoral (IT) delivery [8]. Among these, SC and IT routes are especially promising and have been widely studied to treat cancer with various protein therapeutics [9,10]. However, when it comes to ADCs, there are limited studies that explore their therapeutic potential following SC or IT administration, and there is a need to perform dedicated investigations to evaluate the therapeutic potential of these routes for ADCs.

From the PK/PD perspective, SC and IT routes can potentially improve the therapeutic index of ADCs. SC administration leads to slower absorption of the drug from the site of administration, which results in lower maximum concentration ($C_{max}$) and comparable area under the plasma concentration–time curve (AUC) for the drugs when compared to IV administration [11]. Therefore, SC-administered ADCs may have improved tolerability without compromising the efficacy, if one assumes that $C_{max}$ drives the toxicity and AUC drives the efficacy. Moreover, assuming drug exposure within the tumor drives the efficacy [12], IT administration of ADCs can significantly increase tumor exposure and enhance the efficacy of ADCs. To objectively evaluate these assumptions, here we investigated the PK and PD of an ADC in mice following IV, SC, and IT administration.

SC administration is the most attractive alternative to IV injection, as it provides several advantages such as self-administration, reduced treatment burden, improved patient compliance, reduced infusion-related reactions, and treatment for patients with poor venous access or at risk of venous exhaustion [13,14]. Currently, about 30% of approved mAbs are delivered via SC administration [13,15]. In contrast, the clinical experience with subcutaneously-delivered ADCs is still limited. To the best of our knowledge, there are only two published case studies where SC administration of ADCs in patients has been reported [16,17,18]. First, SC injection of sacituzumab govitecan (IMMU-132) was investigated in patients with different cancer types, including triple-negative breast cancer, metastatic colon cancer, and gastric cancer [18]. The SC dosing regimens of sacituzumab govitecan tested in these case studies were 2~4 mg/kg daily for one week or twice/thrice weekly for two weeks. No significant adverse reaction at the SC injection site was observed, while local erythema that resolved within a few weeks was noted in some patients. Another clinical case study (N = 40, NCT04460456) is reported with SBT6050, an ADC that comprises a toll-like receptor 8 (TLR8) agonist linker-payload conjugated to the HER-targeting mAb pertuzumab [16,17]. The SC dosing regimens of SBT6050 were 0.3~1.2 mg/kg every two weeks as monotherapy or in combination with pembrolizumab. While SC administration of SBT6050 was reported to have a manageable safety profile, the most frequent treatment-emergent adverse events were injection site reactions (50%, 33%, and 3% grade 1, 2, and 3 reactions, respectively) [17]. These clinical observations suggest that it is necessary to evaluate the local toxicity of SC-administered ADCs.

There are also some common challenges associated with the development of SC-administered therapeutic proteins, such as reduced bioavailability, highly concentrated formulation, and immunogenicity [13,15]. In addition, challenges specifically associated with SC delivery of ADCs may include presystemic catabolism and variable bioavailability of different ADC analytes [19,20,21]. Presystemic catabolism can occur at the site of SC injection and during lymphatic transport, and proteolytic enzymes in the SC and lymphatic tissues are responsible for it [20]. Given that >30 protease enzymes (i.e., cathepsin-B) are present in the skin [21], the presence of proteases locally may not only cause the degradation of the mAb component of ADCs but may also lead to the local release of the payload, resulting in ADC-driven toxicity at the injection site. On the other hand, following SC administration of ADCs, different ADC analytes present at the site of injection may demonstrate different rates and extents of absorption into the systemic circulation, which could result in a systemic PK that is different from what is observed following IV administration of ADCs. The altered PK profile may also lead to a different pharmacological profile of the ADC following SC administration. Thus, PK/PD behavior of ADCs can be different following IV and SC administration, and here we investigated this using a preclinical model. Specifically, the PK of different ADC analytes in plasma and tumor, along with the efficacy and local tolerability of the ADC, was evaluated following SC administration. Our goal was to establish the dose–exposure and exposure–response (E-R) relationship for an ADC administered via the SC route using a preclinical model.

Another promising route for ADC administration is the IT route. IT injections have been used for different anticancer modalities, including small molecules, nucleic acids, protein therapeutics, viral vectors, and cell therapies [22]. IT immunotherapy as a neoadjuvant therapy (treatment before surgery) has been demonstrated to improve patient outcomes in early-stage solid tumors [23]. In addition, it is proposed that IT administration may help overcome the resistance against immune checkpoint inhibitors observed following IV injection [24]. Indeed, the techniques for IT injection have recently evolved to enhance the accuracy and precision of drug delivery, using image guidance generated by cross-sectional tomography, ultrasound, and X-ray fluoroscopy [25]. Currently, three biotherapeutics that employ the IT route have been approved to treat cancer, including talimogene laherparepvec (Imlygic^®^), an oncolytic virus for melanoma [26], Teserpaturev/G47Δ (Delytact^®^), an oncolytic virus for glioblastoma [27], and NBTXR3 (Hensify^®^), a radioenhancer nanoparticle for soft-tissue sarcoma [28]. In addition, ~300 clinical trials are ongoing for the evaluation of IT route for anticancer drug administration. While the IT route is being evaluated broadly for various types of immunotherapy agents at preclinical and clinical stages, limited studies have investigated the IT route for ADC administration, most of which come from older references. For example, in the early 90s, Kitamura and colleagues reported that IT injection of an ADC (A7 mAb-neocarzinostatin conjugate) showed significantly better efficacy and lower systemic toxicity than IV injection in mouse xenograft models of pancreatic and colorectal cancer [29,30,31]. However, a thorough evaluation of the IT route as a potential option for ADC administration is still lacking. It also remains to be seen if IT administration can indeed improve the therapeutic index of ADCs. Consequently, here we evaluated the PK of different ADC analytes in plasma and tumor, along with the efficacy of the ADC, following IT administration in a preclinical model. We also established the dose–exposure and E-R relationship for the ADC administered via the IT route.

In order to perform our investigation, trastuzumab, a mAb targeting the human epidermal growth factor receptor 2 (HER2), conjugated with vc-MMAE (T-vc-MMAE), was used as a model ADC [32]. Considering vc-MMAE is one of the most widely used and studied linker-payloads, with four MMAE-based ADCs already in the clinic (brentuximab vedotin, polatuzumab vedotin, tisotumab vedotin, enfortumab vedotin, and disitamab vedotin), T-vc-MMAE serves as a reasonable model ADC [33]. We first investigated the in vivo PK of T-vc-MMAE and different analytes of the ADC in plasma and tumors after IV, SC, and IT administration. Exposures ($C_{max}$ and AUC) of different ADC analytes in plasma and tumors after IT and SC injection were compared with IV injections. Subsequently, the in vivo efficacy of the ADC administered via the three routes was evaluated at different dose levels in an animal model. In addition, the local toxicity of SC-administered ADC was evaluated in both immunocompetent and immunodeficient mice. Finally, a semi-mechanistic PK/PD model was developed using all the PK and efficacy data obtained following IV, SC, and IT administration of the ADC, and the model was further utilized to investigate and optimize clinically evaluated dosing regimens of MMAE-based ADCs for different routes of administration.

## 2. Materials and Methods

### 2.1. Synthesis and Characterization of T-vc-MMAE ADC

The tool ADC, T-vc-MMAE, was synthesized and characterized in-house using our previously published protocol [34]. The average drug-to-antibody ratio (DAR) was confirmed by the hydrophobic interaction chromatography (HIC) method [35,36].

### 2.2. Development of Xenograft Mouse Model

The gastric carcinoma cell NCI-N87 (CRL-5822™) purchased from American Type Tissue Culture (Manassas, VA, USA) was used to develop the xenograft tumors. These cells overexpress HER2, which is the target for T-vc-MMAE. The cells were grown in the RPMI1640 medium (ATCC^®^ 302001™) supplemented with heat-inactivated 10% *v*/*w* fetal bovine serum (Gibco/Thermo Fisher Scientific, Grand Island, NY, USA) and 10 μg/mL of gentamycin (Sigma-Aldrich, St. Louis, MO, USA). Cells were cultured in a humidified incubator maintained with 5% carbon dioxide at 37 °C.

Male athymic nude mice (IMSR_JAX:007850) were purchased at an age of five weeks from the Jackson Laboratory (Bar Harbor, ME, USA). After acclimation to the new conditions for two weeks, mice were subcutaneously injected with NCI-N87 cells into the right dorsal flank. The in vivo study adhered to the Principles of Laboratory Animal Care (National Institutes of Health publication 85–23, revised 1985) and was approved by the University at Buffalo Institutional Animal Care and Use Committee (IACUC #PROTO202100089).

### 2.3. PK of T-vc-MMAE Administered via IT, SC, and IV Route in Tumor-Bearing Mice

The overall in vivo study design is shown in Figure 1. For the PK study, groups of 19 (body weight 24~33 g), 15 (28~34 g), and 15 (26~36 g) mice bearing NCI-N87 xenografts (average 150–200 mm^3^) received 10 mg/kg of T-vc-MMAE single dose via IT, SC, and IV injection, respectively. Blood samples were collected at 10 min and 1, 6, 16, 24, 72, and 168 h for each administration route. Tumor samples were collected at 10 min and 6, 24, 72, and 168 h after the terminal blood sample collection, with three mice sacrificed at each time point, except for the IT dosing group, where five mice were sacrificed at 5 min and 1 h.

### 2.4. Bioanalytical Method

#### 2.4.1. Sample Preparation

A detailed tumor homogenization procedure has been published by us previously [37]. Briefly, RIPA buffer containing protease inhibitor was added to the weighed tumor samples to obtain a dilution factor of 5. Tumor samples were homogenized using a BeadBug™ microtube homogenizer (Benchmark, NJ, USA) at the maximum speed for 15 s, followed by a 30-s ice cool down, and repeated 3–5 times.

#### 2.4.2. ELISA Method to Quantify Total mAb

A sandwich ELISA protocol validated for plasma and tissues was used to quantify the total mAb component of T-vc-MMAE [37]. Briefly, 384-well plates were coated with capturing mAb, anti-human IgG-F(ab’)2 mAb, overnight at 4 °C followed by 1 h blocking at room temperature. Samples, standards, and QCs were loaded and incubated at room temperature for 2 h. Goat anti-human IgG-F(ab’)2 conjugated with alkaline phosphatase was used as the secondary mAb. Absorbance change was measured over time at 405 nm, and all standard curves were fitted using the four-parameter logistic equation.

#### 2.4.3. LC-MS/MS to Quantify Unconjugated MMAE

The Shimadzu 8040 LC-MS/MS system with electrospray ionization and triple quadrupole mass spectrometer was used. The XBridge BEH Amide column (Waters, Milford, MA, USA) was used with the aqueous phase as water and the organic phase as 95:5 acetonitrile/water (both with 5 mM ammonium formate and 0.1% formic acid). The duration of the chromatographic run was 12 min, where two MRM scans (718.5/686.5 and 718.5/152.1 amu) were monitored. Deuterated (d8) MMAE (MCE MedChem Express, NJ, USA) was used as an internal standard. The samples (standard, QC, plasma, and tumor homogenates) were spiked with d8-MMAE, and acetonitrile was added followed by the vortexing, centrifugation (15,000× *g* for 15 min at 4 °C), and collection of supernatants. The supernatants were evaporated and reconstituted with 50 µL acetonitrile/water (95:5 *v*/*v*) containing 0.1% formic acid.

#### 2.4.4. Papain Deconjugation Method to Quantify Total MMAE

The cysteine protease papain (Sigma-Aldrich) was used to cleave the MMAE from the linker and release the payload from ADC, which enabled quantification of total MMAE (conjugated MMAE + unconjugated MMAE). Plasma and tumor samples were digested by incubating with papain for 8 h at 37 °C. After incubation, samples were treated as described in the unconjugated MMAE section for total MMAE quantification.

### 2.5. Efficacy Study of T-vc-MMAE Administered via IT, SC, and IV Route in Tumor-Bearing Mice

The overall in vivo efficacy study design is shown in Figure 1. A total of 59 NCI-N87 tumor-bearing xenografts were divided into 4 groups, receiving IT (N = 20), SC (N = 18), or IV (N = 15) administration of T-vc-MMAE single dose, or as the control group (N = 6). For each administration route, mice were divided into three groups treated with either high, mid, or low doses of T-vc-MMAE. The injection volumes were 8 µL per mouse for IT administration and 100 µL per mouse for SC dosing. The sample size and initial tumor volume (mean and SD) for each route of administration and different dose levels are summarized in Appendix A. Tumor volume and body weight were monitored three times per week for ~110 days or until tumor volume reached >2000 mm^3^. Tumor volumes were calculated by measuring the length and width of tumors using a vernier caliper and calculated using the formula of $1/2\cdot length\cdot {width}^{2}$. The selection of the dose for the high-, mid-, and low-dose groups for each route of administration was based on the results of the PK study, where the goal was to achieve similar systemic exposure of the ADC for each administration route at each dose level. The tumor growth curves are presented as mean and SD calculated from the observed data pooled in bins, as well as individual data from each mouse.

### 2.6. Local Toxicity Study of SC Administered T-vc-MMAE in Mice

The design of the local toxicity study for SC-administered ADC is shown in Figure 1. Two mouse strains, including immunocompetent C57BL/6J WT mice (IMSR_JAX:000664, the Jackson Laboratory) and immunodeficient athymic nude mice, were used for the investigation. For each strain of mouse, a total of 12 mice were divided into four groups (N = three per group), receiving 30 mg/kg of naked mAb (Herceptin^®^) at a single dose, 0.5 mg/kg of naked MMAE at a single dose, 30 mg/kg of T-vc-MMAE at a single dose, or vehicle control (DPBS), subcutaneously. The SC injection volume was 100 µL for each mouse. After 96 h of SC dosing, all animals were euthanized, and the full-thickness skin tissue surrounding the SC injection site was excised. The skin tissues underwent fixation and paraffin processing by sectioning at 10 µm and staining with hematoxylin and eosin for histopathology analysis.

The histology slides from each mouse (five slide sections per mouse) were examined by experienced pathologists, who were blinded to the experiment. Semiquantitative analysis was performed, where tissue damages, including blister formation, inflammatory infiltrate, ulceration, and necrosis, were recorded and scored. A score of 0 to 3 indicated normal, mild, moderate, and severe damage, respectively. Values from each mouse were summed up to obtain a total tissue damage score. Statistical analysis was performed to compare total tissue damage scores obtained from different groups using an unpaired *t*-test. A *p*-value < 0.05 was considered to be statistically significant.

### 2.7. Development of a PK/PD Model for T-vc-MMAE Administered via IT, SC, and IV Routes

#### 2.7.1. PK Model Structure

The structure of the final semi-mechanistic PK model is shown in Figure 2. Table 1 provides detailed descriptions of the model parameter symbols, units, and the source of model parameters. PK/PD model equations are provided below. PK of ADC in plasma is described using Equations (1)–(9), and PK of ADC in tumors is described using Equations (10)–(19). Equations for the PD model are shown in Equations (20)–(24).

PK/PD model equations:(1)$$\frac{d{X1}_{ADC}}{dt}=-{\mathrm{CLD}}_{mAb}\times \frac{{X1}_{ADC}}{{V1}_{mAb}}+{\mathrm{CLD}}_{mAb}\times \frac{{X2}_{ADC}}{{V2}_{mAb}}-{\mathrm{CL}}_{mAb}\times \frac{{X1}_{ADC}}{{V1}_{mAb}}-k_{dec,ADC}\times {X1}_{ADC}\;\;\;\;\;-\left(\frac{2\times P_{ADC}\times R_{cap}}{R_{krough}^{2}}\right)\times \left(\frac{{X1}_{ADC}}{{V1}_{mAb}}-\frac{X_{ADC_{unbound}}^{tumor}/V_{tumor}}{\varepsilon_{ADC}}\right)\times V_{tumor}\;\;\;\;\;\;\;\;-\left(\frac{6\times D_{ADC}}{R_{tumor}^{2}}\right)\times \left(\frac{{X1}_{ADC}}{{V1}_{mAb}}-\frac{X_{ADC_{unbound}}^{tumor}/V_{tumor}}{\varepsilon_{ADC}}\right)\times V_{tumor}+ka\times {Xa}_{ADC}$$
(2)$$\frac{d{X2}_{ADC}}{dt}={\mathrm{CLD}}_{mAb}\times \frac{{X1}_{ADC}}{{V1}_{mAb}}-{\mathrm{CLD}}_{mAb}\times \frac{{X2}_{ADC}}{{V2}_{mAb}}-k_{dec,ADC}\times {X2}_{ADC}$$
(3)$$\frac{d{X1}_{mAb}}{dt}=-{\mathrm{CLD}}_{mAb}\times \frac{{X1}_{mAb}}{{V1}_{mAb}}+{\mathrm{CLD}}_{mAb}\times \frac{{X2}_{mAb}}{{V2}_{mAb}}-{\mathrm{CL}}_{mAb}\times \frac{{X1}_{mAb}}{{V1}_{mAb}}+k_{dec,ADC}\times {X1}_{mAb}\;\;\;\;-\left(\frac{2\times P_{ADC}\times R_{cap}}{R_{krough}^{2}}\right)\times \left(\frac{{X1}_{mAb}}{{V1}_{mAb}}-\frac{X_{mAb\_unbound}^{tumor}/V_{tumor}}{\varepsilon_{ADC}}\right)\times V_{tumor}\;-\left(\frac{6\times D_{ADC}}{R_{tumor}^{2}}\right)\times \left(\frac{{X1}_{ADC}}{{V1}_{mAb}}-\frac{X_{mAb\_unbound}^{tumor}/V_{tumor}}{\varepsilon_{ADC}}\right)\times V_{tumor}$$
(4)$$\frac{d{X2}_{mAb}}{dt}={\mathrm{CLD}}_{mAb}\times \frac{{X1}_{mAb}}{{V1}_{mAb}}-{\mathrm{CLD}}_{mAb}\times \frac{{X2}_{mAb}}{{V2}_{mAb}}+k_{dec,ADC}\times {X2}_{mAb}$$
(5)$$\frac{d{X1}_{acPL}}{dt}=-{\mathrm{CLD}}_{mAb}\times \frac{{X1}_{acPL}}{{V1}_{mAb}}+{\mathrm{CLD}}_{mAb}\times \frac{{X2}_{acPL}}{{V2}_{mAb}}-{\mathrm{CL}}_{mAb}\times \frac{{X1}_{ADC}}{{V1}_{mAb}}\times DAR-k_{dec,PL}\times {X1}_{acPL}\;-\left(\frac{2\times P_{ADC}\times R_{cap}}{R_{krough}^{2}}\right)\times \left(\frac{{X1}_{acPL}}{{V1}_{mAb}}-\frac{X_{acPL\_unbound}^{tumor}/V_{tumor}}{\varepsilon_{ADC}}\right)\times V_{tumor}\;\;\;\;-\left(\frac{6\times D_{ADC}}{R_{tumor}^{2}}\right)\times \left(\frac{{X1}_{acPL}}{{V1}_{mAb}}-\frac{X_{acPL\_unbound}^{tumor}/V_{tumor}}{\varepsilon_{ADC}}\right)\times V_{tumor}+ka\times {Xa}_{acPL}$$
(6)$$\frac{d{X2}_{acPL}}{dt}={\mathrm{CLD}}_{mAb}\times \frac{{X1}_{acPL}}{{V1}_{mAb}}-{\mathrm{CLD}}_{mAb}\times \frac{{X2}_{acPL}}{{V2}_{mAb}}-k_{dec,PL}\times {X2}_{acPL}$$
(7)$$\frac{d{X1}_{PL}}{dt}=-{\mathrm{CLD}}_{PL}\times \frac{{X1}_{PL}}{{V1}_{PL}}+{\mathrm{CLD}}_{PL}\times \frac{{X2}_{PL}}{{V2}_{PL}}-{\mathrm{CL}}_{PL}\times \frac{{X1}_{PL}}{{V1}_{PL}}+{\mathrm{CL}}_{mAb}\times \frac{{X1}_{ADC}}{{V1}_{ADC}}\times DAR+k_{dec,PL}\times {X1}_{acPL}\;\;\;-\left(\frac{2\times P_{PL}\times R_{cap}}{R_{krough}^{2}}\right)\times \left(\frac{{X1}_{PL}}{{V1}_{PL}}-\frac{X_{PL\_unbound}^{tumor}/V_{tumor}}{\varepsilon_{PL}}\right)\times V_{tumor}\;-\left(\frac{6\times D_{PL}}{R_{tumor}^{2}}\right)\times \left(\frac{{X1}_{PL}}{{V1}_{PL}}-\frac{X_{PL\_unbound}^{tumor}/V_{tumor}}{\varepsilon_{PL}}\right)\times V_{tumor}$$
(8)$$\frac{d{X2}_{PL}}{dt}={\mathrm{CLD}}_{PL}\times \frac{{X1}_{PL}}{{V1}_{PL}}-{\mathrm{CLD}}_{PL}\times \frac{{X2}_{PL}}{{V2}_{PL}}+k_{dec,PL}\times {X2}_{PL}$$
(9)$$DAR=\frac{{X1}_{acPL}}{{X1}_{ADC}+{X1}_{mAb}}$$
(10)$$\begin{array}{cc} \frac{dX_{ADC_{unbound}}^{tumor}}{dt}= & \left(\frac{2\times P_{ADC}\times R_{cap}}{R_{krough}^{2}}\right)\times \left(\frac{{X1}_{ADC}}{{V1}_{mAb}}-\frac{X_{ADC_{unbound}}^{tumor}/V_{tumor}}{\varepsilon_{ADC}}\right)\times V_{tumor}\\ & +\left(\frac{6\times D_{ADC}}{R_{tumor}^{2}}\right)\times \left(\frac{{X1}_{ADC}}{{V1}_{mAb}}-\frac{X_{ADC_{unbound}}^{tumor}/V_{tumor}}{\varepsilon_{ADC}}\right)\times V_{tumor}\\ & -{k_{on}}_{Antigen}^{ADC}\times \frac{X_{ADC_{unbound}}^{tumor}/V_{tumor}}{\varepsilon_{ADC}}\times \left(Antigen-X_{ADC_{bound}}^{tumor}/V_{tumor}\right)\times V_{tumor}\\ & +{k_{off}}_{Antigen}^{ADC}\times X_{ADC_{bound}}^{tumor}-f_{dec}\times k_{dec,ADC}\times X_{ADC_{unbound}}^{tumor} \end{array}$$
(11)$$\frac{dX_{ADC_{bound}}^{tumor}}{dt}={k_{on}}_{Antigen}^{ADC}\times \frac{X_{ADC_{unbound}}^{tumor}/V_{tumor}}{\varepsilon_{ADC}}\times \left(Antigen-X_{ADC_{bound}}^{tumor}/V_{tumor}\right)\times V_{tumor}\;\;\;\;\;\;\;\;\;-{k_{off}}_{Antigen}^{ADC}\times X_{ADC_{bound}}^{tumor}-k_{int\_ADC}\times X_{ADC_{bound}}^{tumor}-f_{dec}\times k_{dec,ADC}\times X_{ADC_{bound}}^{tumor}$$
(12)$$\begin{array}{cc} \frac{dX_{{mAb}_{unbound}}^{tumor}}{dt}= & \left(\frac{2\times P_{ADC}\times R_{cap}}{R_{krough}^{2}}\right)\times \left(\frac{{X1}_{mAb}}{{V1}_{mAb}}-\frac{X_{{mAb}_{unbound}}^{tumor}/V_{tumor}}{\varepsilon_{ADC}}\right)\times V_{tumor}\\ & +\left(\frac{6\times D_{ADC}}{R_{tumor}^{2}}\right)\times \left(\frac{{X1}_{mAb}}{{V1}_{mAb}}-\frac{X_{{mAb}_{unbound}}^{tumor}/V_{tumor}}{\varepsilon_{ADC}}\right)\times V_{tumor}\\ & -{k_{on}}_{Antigen}^{ADC}\times \frac{X_{{mAb}_{unbound}}^{tumor}/V_{tumor}}{\varepsilon_{ADC}}\times \left(Antigen-X_{{mAb}_{unbound}}^{tumor}/V_{tumor}\right)\times V_{tumor}\\ & +{k_{off}}_{Antigen}^{ADC}\times X_{{mAb}_{unbound}}^{tumor}+f_{dec}\times k_{dec,ADC}\times X_{ADC_{unbound}}^{tumor} \end{array}$$
(13)$$\frac{dX_{{mAb}_{bound}}^{tumor}}{dt}={k_{on}}_{Antigen}^{ADC}\times \frac{X_{{mAb}_{bound}}^{tumor}/V_{tumor}}{\varepsilon_{ADC}}\times \left(Antigen-X_{ADC_{bound}}^{tumor}/V_{tumor}\right)\times V_{tumor}\;\;\;\;\;\;\;\;\;-{k_{off}}_{Antigen}^{ADC}\times X_{{mAb}_{bound}}^{tumor}-k_{int\_mAb}\times X_{ADC_{bound}}^{tumor}+f_{dec}\times k_{dec,ADC}\times X_{ADC_{bound}}^{tumor}$$
(14)$$\begin{array}{cc} \frac{dX_{{acPL}_{unbound}}^{tumor}}{dt}= & \left(\frac{2\times P_{ADC}\times R_{cap}}{R_{krough}^{2}}\right)\times \left(\frac{{X1}_{acPL}}{{V1}_{mAb}}-\frac{X_{{acPL}_{unbound}}^{tumor}/V_{tumor}}{\varepsilon_{ADC}}\right)\times V_{tumor}\\ & +\left(\frac{6\times D_{ADC}}{R_{tumor}^{2}}\right)\times \left(\frac{{X1}_{acPL}}{{V1}_{mAb}}-\frac{X_{{acPL}_{unbound}}^{tumor}/V_{tumor}}{\varepsilon_{ADC}}\right)\times V_{tumor}\\ & -{k_{on}}_{Antigen}^{ADC}\times \frac{X_{{acPL}_{unbound}}^{tumor}/V_{tumor}}{\varepsilon_{ADC}}\times \left(Antigen-X_{{acPL}_{unbound}}^{tumor}/V_{tumor}\right)\times V_{tumor}\\ & +{k_{off}}_{Antigen}^{ADC}\times X_{{acPL}_{unbound}}^{tumor}-f_{dec}\times k_{dec,PL}\times X_{{acPL}_{unbound}}^{tumor} \end{array}$$
(15)$$\frac{dX_{{acPL}_{bound}}^{tumor}}{dt}={k_{on}}_{Antigen}^{ADC}\times \frac{X_{{acPL}_{unbound}}^{tumor}/V_{tumor}}{\varepsilon_{ADC}}\times \left(Antigen-X_{{acPL}_{unbound}}^{tumor}/V_{tumor}\right)\times V_{tumor}\;\;\;\;\;\;\;\;\;-{k_{off}}_{Antigen}^{ADC}\times X_{{acPL}_{unbound}}^{tumor}-k_{int\_ADC}\times X_{{acPL}_{bound}}^{tumor}-f_{dec}\times k_{dec,PL}\times X_{{acPL}_{bound}}^{tumor}$$
(16)$$\frac{dX_{{PL}_{extra}}^{tumor}}{dt}=\left(\frac{2\times P_{MMAE}\times R_{cap}}{R_{krough}^{2}}\right)\times \left(\frac{{X1}_{PL}}{{V1}_{PL}}-\frac{X_{{PL}_{extra}}^{tumor}/V_{tumor}}{\varepsilon_{MMAE}}\right)\times V_{tumor}\;\;\;\;\;\;\;\;\;+\left(\frac{6\times D_{MMAE}}{R_{tumor}^{2}}\right)\times \left(\frac{{X1}_{PL}}{{V1}_{PL}}-\frac{X_{{PL}_{extra}}^{tumor}/V_{tumor}}{\varepsilon_{MMAE}}\right)\times V_{tumor}-k_{in}\times X_{{PL}_{extra}}^{tumor}\;\;\;\;\;\;+k_{out}\times X_{{PL}_{intra\_unbound}}^{tumor}-f_{dec}\times k_{dec,PL}\times \left(X_{{acPL}_{bound}}^{tumor}+X_{{acPL}_{unbound}}^{tumor}\right)$$
(17)$$\begin{array}{cc} \frac{dX_{{PL}_{intra\_unbound}}^{tumor}}{dt} & =k_{in}\times X_{{PL}_{extra}}^{tumor}-k_{out}\times X_{{PL}_{intra}}^{tumor}-{k_{on}}_{Tubulin}^{MMAE}\times \frac{X_{{PL}_{intra\_unbound}}^{tumor}}{V_{tumor}}\times (Tubulin\\ & -\frac{X_{{PL}_{intra\_bound}}^{tumor}}{V_{tumor}})\times V_{tumor}+{k_{off}}_{Tubulin}^{MMAE}\times X_{{PL}_{intra\_bound}}^{tumor}+k_{int}^{ADC}\times X_{{acPL}_{bound}}^{tumor}\times {DAR}_{tumor} \end{array}$$
(18)$$\begin{array}{cc} \frac{dX_{{PL}_{intra\_bound}}^{tumor}}{dt}= & {k_{on}}_{Tubulin}^{MMAE}\times \frac{X_{{PL}_{intra\_unbound}}^{tumor}}{V_{tumor}}\times (Tubulin-\frac{X_{{PL}_{intra\_bound}}^{tumor}}{V_{tumor}})\times V_{tumor}\\ & -{k_{off}}_{Tubulin}^{MMAE}\times X_{{PL}_{intra\_bound}}^{tumor} \end{array}$$
(19)$${DAR}_{tumor}=\frac{X_{{acPL}_{unbound}}^{tumor}+X_{{acPL}_{bound}}^{tumor}}{X_{ADC_{unbound}}^{tumor}+X_{ADC_{bound}}^{tumor}+X_{{mAb}_{unbound}}^{tumor}+X_{{mAb}_{bound}}^{tumor}}$$
(20)$$\%TE=\frac{X_{{PL}_{intra\_bound}}^{tumor}}{Tubulin}\times 100\%$$
(21)$$\frac{d{TV}_{1}}{dt}=\frac{2\times k_{ge}\times k_{gl}\times {TV}_{1}\times \left({TV}_{1}/V_{tumor}\right)}{k_{ge}+2\times k_{ge}\times {TV}_{1}}-\frac{K_{max}\times \%TE}{{EC}_{50}+\%TE}\times {TV}_{1}$$
(22)$$\frac{d{TV}_{2}}{dt}=\frac{K_{max}\times \%TE}{{EC}_{50}+\%TE}\times {TV}_{1}-\frac{{TV}_{2}}{tau}$$
(23)$$\frac{d{TV}_{3}}{dt}=\frac{\left({TV}_{2}-{TV}_{3}\right)}{tau}$$
(24)$$\frac{d{TV}_{4}}{dt}=\frac{\left({TV}_{3}-{TV}_{4}\right)}{tau}$$

The model can simultaneously characterize the PK of conjugates (conjugated mAb, conjugated MMAE), unconjugated MMAE, and naked mAb in plasma and tumor, following T-vc-MMAE administration via the SC, IV, and IT routes. The left side of Figure 2 shows the PK model structure used to describe plasma PK of ADC. When each conjugate molecule undergoes catabolism (characterized by ${CL}_{mAb}$), it is assumed to release a certain number of payloads equivalent to the DAR value at the given time (${CL}_{mAb}\times DAR)$. Additionally, the conjugated MMAE can deconjugate from the ADC ($k_{dec,PL})$ and contribute payload molecules to the free MMAE compartment. The formation of naked mAb, once conjugated mAb releases all its conjugated payloads, is characterized using $k_{dec,ADC}$. We have previously observed that the conjugation of vc-MMAE at DAR ~4 does not affect the PK of mAb in NCI-N87 tumor-bearing mice. Therefore, the PK of conjugated mAb and naked mAb in the plasma and peripheral compartments was described using the same two-compartmental PK model (parameterized in terms of ${CL}_{mAb}$, ${CLD}_{mAb}$, ${V1}_{mAb}$, and ${V2}_{mAb}$).

The right side of Figure 2 shows the tumor disposition model for the ADC [38,39]. In the tumor model, each ADC analyte (conjugated mAb, naked mAb, conjugated MMAE, and free MMAE) is allowed to move between plasma and tumor extracellular space via vascular exchange (extravasation) and surface exchange (diffusion) pathways. These pathways are defined by the permeability and diffusion coefficient of the molecules. Both pathways also depend on vascular density and tumor size. Conjugated mAb and naked mAb within the tumor extracellular space can bind to the target antigen and be internalized into the tumor cell. After internalization, these molecules undergo degradation inside the tumor cell and release the payload equivalent to the DAR at the given time ($k_{int}\times DAR$). The unbound or antigen-bound conjugates within the extracellular space can undergo deconjugation and release free payloads. Since the activity and expression of proteases (i.e., cathepsin B) are reported to be significantly higher inside the tumor microenvironment, the deconjugation process of ADC is assumed to be higher in the tumor microenvironment than in the systemic circulation. Thus, the deconjugation rate in the tumor is multiplied by a factor ($f_{dec}$). The released payloads inside the tumor cell can bind to the intracellular target (i.e., tubulin for MMAE) or efflux out of the tumor cell. On the other hand, the free payload within the extracellular space can diffuse into the tumor cell and bind to the intracellular target.

The SC absorption of T-vc-MMAE is described by a first-order absorption rate ($k_{a}$) and a bioavailability parameter (F). During IT injection, ADC is instantly and directly delivered into the tumor extracellular space. However, since the tumors can only hold up a certain volume of IT injection, the amount of dose retained in the tumor compartment is described by a relative bioavailability term, $F_{IT}$. Any injected volume that does not stay in the tumors is assumed to be immediately released into the plasma.

**Table 1 pharmaceutics-15-01132-t001:** A list of literature-derived or model-estimated parameters used by the semi-mechanistic PK/PD model.

Parameter	Definition	Value	Unit	Source
Plasma PK parameters				
${\mathrm{CL}}_{mAb}$	Plasma clearance of ADC/mAb	0.006 (10.8%)	mL/h	Estimated ^1^
${\mathrm{CLD}}_{mAb}$	Distribution clearance of ADC/mAb	0.040 (28.9%)	mL/h	Estimated ^1^
${V1}_{mAb}$	Volume of distribution of ADC/mAb in the central compartment	1.41 (6.48%)	mL	Estimated ^1^
${V2}_{mAb}$	Volume of distribution of ADC/mAb in the peripheral compartment	0.861 (13.3%)	mL	Estimated ^1^
${\mathrm{CL}}_{PL}$	Plasma clearance of unconjugated MMAE	15.9	mL/h	Estimated ^2^
${\mathrm{CLD}}_{PL}$	Distribution clearance of unconjugated MMAE	0.811 (20.0%)	mL/h	Estimated ^2^
${V1}_{PL}$	Volume of distribution of unconjugated MMAE in the central compartment	2.25 (9.68%)	mL	Estimated ^2^
${V2}_{PL}$	Volume of distribution of ADC/mAb in the central compartment	5.60 (25.7%)	mL	Estimated ^2^
$k_{dec,ADC}$	Deconjugation rate constant of ADC to form naked mAb	0.00344 (9.65%)	1/h	Estimated
$k_{dec,PL}$	Deconjugation rate constant of ADC to release MMAE	0.00905 (6.48%)	1/h	Estimated
Tumor PK parameters				
$P_{ADC}$	Permeability of ADC across tumor blood vessels	0.01	mm/h	[40]
$P_{MMAE}$	Permeability of MMAE across tumor blood vessels	0.0875	mm/h	[41]
$D_{ADC}$	Diffusion rate of ADC	0.00054	mm^2^/h	[42]
$D_{MMAE}$	Diffusion rate of MMAE	1.04	mm^2^/h	[41]
$R_{cap}$	Tumor blood capillary radius	0.008	mm	[43]
$R_{krough}$	The average distance between two capillaries	0.075	mm	[43]
TV0	Initial tumor volume	measured	mm^3^	Experimental
$R_{tumor}$	Tumor radius	dynamic	mm	Derived ^3^
$\varepsilon_{ADC}$	Tumor void volume for ADC	0.24	-	[41]
$\varepsilon_{MMAE}$	Tumor void volume for MMAE	0.44	-	[41]
${k_{on}}_{Antigen}^{ADC}$	Association rate constant between ADC and HER2	1.25	1/nM/h	[44]
${k_{off}}_{Antigen}^{ADC}$	Dissociation rate constant between ADC and HER2	2.26	1/h	[44]
$k_{int\_ADC}$	Internalization rate of ADC-antigen complex inside the cell	0.112 (19.6%)	1/h	Estimated
$k_{int\_mAb}$	Internalization rate of mAb-antigen complex inside the cell	0.027	1/h	[45]
Antigen	Total antigen concentration	1799	nM	[44]
${k_{on}}_{Tubulin}^{MMAE}$	Secondary order association rate constant between MMAE and tubulin	0.00187	1/nM/h	[46,47]
${k_{off}}_{Tubulin}^{MMAE}$	First-order dissociation rate constant between MMAE-tubulin complex	0.545	1/h	[38,46]
Tubulin	Total tubulin concentration	500	nM	[48]
$k_{in}$	MMAE nonspecific uptake rate in cancer cell	0.075	1/h	[49]
$k_{out}$	MMAE efflux rate from the cell	0.0116 (30.9%)	1/h	Estimated
$f_{dec}$	Fold-increase of deconjugation rate in tumor	30.8 (20.7%)	-	Estimated
Route-specific PK parameters				
$F_{IT}$	Percentage of injected dose retained in tumor after IT administration	75.0 (4.61%)	%	Estimated
F	Bioavailability for SC administered ADC	47.6 (0.836%)	%	Estimated
ka	Absorption rate constant for SC administered ADC	0.0498 (1.96%)	1/h	Estimated
PD parameters				
$k_{gl}$	Zero-order rate constant of tumor growth	3.08 (24.7%)	mm^3^/h	Estimated ^4^
$k_{ge}$	First-order rate constant of tumor growth	0.0018 (3.21%)	1/h	Estimated ^4^
$K_{max}$	Maximum cell killing rate	0.00673 (6.18%)	1/h	Estimated
${IC}_{50}$	Percentage of tubulin occupied by MMAE that produces 50% of $k_{\max}$	17.9 (1.17%)	%	Estimated
tau	Mean transit time for the cell distribution model	19.3 (21.6%)	h	Estimated
${Var\_k}_{gl}$	Random effect population variability for tumor growth in the linear phase	53.8 (17.2%)	%	Estimated ^4^
${Var\_K}_{max}$	Random effect population variability for maximum tumor killing rate	43.2 (4.93%)	%	Estimated
Var_tau	Random effect variability for transit time between different administration routes	131 (16.3%)	%	Estimated

^1^ PK parameters were estimated by fitting plasma PK data of trastuzumab in mice obtained from [37] to a two-compartmental model. ^2^ PK parameters were estimated by fitting plasma PK data of free MMAE in mice obtained from [47] to a two-compartmental model, and CL_PL_ was obtained by non-compartmental analysis. ^3^ Parameter calculated based on tumor volume of individual mice observed during the experimental period. ^4^ Tumor growth parameters were estimated using control group tumor volume data.

#### 2.7.2. PD Model Structure

The PK model for T-vc-MMAE is connected to a PD model as shown in Figure 2. Equations of the PD model are shown in Equations (20)–(24), and descriptions of the model parameter symbols, units, and the source of the parameters are provided in Table 1. The final PK/PD model is used to characterize all the efficacy data for T-vc-MMAE observed following the administration of ADC via SC, IV, and IT routes. The percentage of tubulin occupied by MMAE, represented as intracellular target engagement (%TE) predicted by the PK model, is used to drive the efficacy of T-vc-MMAE using a nonlinear killing function. It is assumed that the antitumor activity of T-vc-MMAE leads to a portion of tumor cells switching from proliferating to a nonproliferating state, which eventually leads to cell death. Thus, a transit compartment model is incorporated to account for the delay of the ADC treatment effect, and the residence time of cells in each transit compartment is characterized using the parameter tau. The growth of NCI-N87 tumor cells in untreated mice is characterized using an initial exponential tumor growth phase followed by a linear growth phase as the tumor volume increases.

#### 2.7.3. PK/PD Model Fitting

The PK/PD model development process for T-vc-MMAE administered via SC, IV, and IT routes included five steps. In step 1, our previously reported plasma PK data for naked mAb in nude mice [37] was used to estimate the PK parameters of the mAb component of T-vc-MMAE, by fitting a two-compartmental model to the data. In step 2, previously reported plasma PK data of naked MMAE in nude mice [47] were used to estimate the PK parameters of unconjugated MMAE in plasma. It was assumed that unconjugated MMAE behaves similarly to the naked MMAE administrated in the free form. PK parameters estimated in steps 1 and 2 were then fixed in the following steps. In step 3, parameters for the tumor disposition model obtained from various literature (Table 1) were incorporated and fixed in the semi-mechanistic PK model of T-vc-MMAE. Plasma and tumor concentration–time data for T-vc-MMAE obtained from IV administration were used to estimate $k_{dec}$, $k_{int}$, and $f_{dec}$ parameters; data for IT injection were used to estimate $F_{IT}$; and data for SC administration were used to estimate F and $k_{a}$. In step 4, tumor growth data from untreated mice were used to estimate the rate of exponential ($k_{ge}$) and linear ($k_{gl}$) tumor growth (interindividual variability assigned to $k_{gl}$), which were then fixed for the subsequent PD model development. In step 5, the PK model established in steps 1–4 was used to predict %TE inside tumor cells, which served as the driving force for ADC cytotoxicity in the PD model. Tumor growth inhibition data from all dose levels of the three routes were used to estimate parameters for the nonlinear killing function ($K_{max},{EC}_{50}$) and tau, where the population variability was assigned to $K_{max}$ and tau. Since the confidence in the measurement of smaller tumor volumes (i.e., <50 mm^3^) by the caliper is low, tumor volumes <50 mm^3^ were treated as interval-censored data for the modeling.

### 2.8. PK/PD Model Simulation

The semi-mechanistic PK/PD model for T-vc-MMAE was used to objectively evaluate the therapeutic potentials of T-vc-MMAE administrated via SC, IT, and IV routes through simulations of tumor growth inhibition (TGI) profiles in various scenarios. First, we simulated TGI using clinically approved dosing regimens of MMAE-based ADCs (i.e., 1.8 mg/kg every three weeks, Q3W) when treated via IV, SC, or IT routes for six cycles. Then, we examined what dose levels are required for IT and SC routes to achieve comparable TGI as IV injection when keeping the dosing frequency the same (Q3W). We also explored what dosing frequencies were needed for IT and SC routes to attain similar TGI as IV routes when treating with the same dose amounts (i.e., 1.8 mg/kg).

### 2.9. Data Analysis

Noncompartmental analysis (NCA) was conducted for plasma and tumor PK data. AUC computed from time 0 to the last observed time (${AUC}_{0-t}$) was calculated using the linear/log trapezoidal method in WinNonlin (version 8.1, Pharsight, St. Louis, MO, USA). The PK model was fitted to the PK data using the maximum likelihood estimation method in the ADAPT software version 5 (Biomedical Simulations Resource, University of Southern California, Los Angeles, CA, USA), assuming an additive plus proportional error variance model. The Monolix software (2021R2, Lixoft SAS, a Simulations Plus company, CA, USA) was used to estimate PD parameters using the stochastic approximation expectation maximization (SAEM) algorithm. Statistical analysis for the local toxicity data was conducted using GraphPad Prism (unpaired *t*-test), where *p* < 0.05 was considered to be significant.

## 3. Results

### 3.1. Synthesis and Characterization of T-vc-MMAE ADC

Appendix A provides the HIC profiles for T-vc-MMAE and the parent mAb (i.e., trastuzumab). The calculated average DAR value was ~3.5.

### 3.2. Bioanalytical Method Development

For the sandwich ELISA, the lower limit of quantification (LLOQ) was 1 ng/mL, and the upper limit of quantification (ULOQ) was 500 ng/mL for plasma and tumor samples. The QCs for plasma and tumor were within ±20% of the nominal value. For the LC/MS/MS method, the LLOQ of MMAE was 0.025 ng/mL for plasma and tumor samples. The QCs for plasma and tumor were within ±15% of the nominal value.

### 3.3. In Vivo PK of T-vc-MMAE Administered via IT, SC, and IV Routes

Figure 3 shows the plasma PK profiles of total mAb (Figure 3a), total MMAE (Figure 3b), free MMAE (Figure 3c), and conjugated MMAE (calculated as total MMAE—free MMAE) (Figure 3d) observed after IT, SC, and IV administration of a single 10 mg/kg dose of T-vc-MMAE. Figure 4 shows the tumor PK profiles of total mAb (Figure 4a), total MMAE (Figure 4b), free MMAE (Figure 4c), and conjugated MMAE (Figure 4d) after IT, SC, and IV administration of single 10 mg/kg dose of T-vc-MMAE. Appendix A show the plasma PK profiles superimposed over the tumor PK profiles of different ADC analytes after IT, SC, and IV administration, respectively. $C_{max}$, ${AUC}_{0-t}$, and $AUC_{inf}$ values for multiple analytes of T-vc-MMAE in plasma and tumor calculated using NCA are shown in Table 2. Plasma $C_{max}$ and ${AUC}_{0-t}$ values for different analytes of ADC obtained after IT, SC, and IV administration were compared and represented as percentage ratio (%) values in Table 2. Additionally, tumor to plasma ${AUC}_{0-t}$ ratios (%) for IV, IT, and SC routes are also shown in Table 2.

#### 3.3.1. Plasma PK

A prolonged exposure of total mAb was observed after IT, IV, and SC injection of ADC, with a half-life ($T_{1/2}$) of about 27 days for all three routes. On the other hand, the $T_{1/2}$ of total MMAE, unconjugated MMAE, and conjugated MMAE were about three days for IT, IV, and SC administrations. The shorter $T_{1/2}$ of total MMAE, conjugated MMAE, and unconjugated MMAE compared to total mAb in plasma suggests deconjugation and fast elimination of released payload upon deconjugation. Since $T_{1/2}$ values for the three routes were similar, it indicates that different administration routes did not affect the PK behavior of the ADC. Concentration–time profiles of conjugated MMAE were derived by subtracting free MMAE concentrations from total MMAE concentrations. The PK profiles of total MMAE and conjugated MMAE were similar, indicating that conjugated MMAE primarily determines the exposure of total MMAE, and unconjugated MMAE has minimal contribution towards total MMAE concentrations.

After SC injection, $T_{max}$ in plasma for all ADC analytes occurred at 24 h. After IT injection, $T_{max}$ in plasma was 24 h for total mAb, 6 h for total MMAE and conjugated MMAE, and 10 min for unconjugated MMAE (Table 2). Following IT administration, plasma concentrations showed relatively high variability at early time points, while the variability became less at the later time points (Figure 3). CV% for maximum concentrations ($C_{max}$) in plasma for all analytes were higher for IT (18%~44%) compared to SC and IV administration (<5%) (Table 2).

After IT administration, $C_{max}$ in plasma for all ADC analytes decreased by >50% compared to IV administration, with free MMAE decreasing the most (about 75% decrease). Regarding systemic exposures, ${AUC}_{0-t}$ after IT injection were almost identical to IV injection for total mAb, total MMAE, and conjugated MMAE, with ${AUC}_{0-t}$ ratios of IT to IV about 99%. Importantly, IT administration resulted in lower systemic exposure of unconjugated MMAE compared to the IV route, with ${AUC}_{0-t}$ ratio of about 70%. It is known that free MMAE in systemic circulation may cause off-target toxicity. Therefore, the lower $C_{max}$ and AUC of unconjugated MMAE in plasma observed after IT injection suggests that IT route may potentially reduce the toxicity of MMAE-based ADCs.

SC administration of T-vc-MMAE resulted in a >70% decrease in plasma $C_{max}$ for all ADC analytes, with unconjugated MMAE decreasing the most (86%). It is assumed that $C_{max}$ in plasma is related to toxicity. Therefore, SC injection of ADCs enables reduced plasma $C_{max}$ and may improve the safety profile of ADCs. Plasma ${AUC}_{0-t}$ after SC administration were about 50~80% compared to IV administration, depending on different ADC analytes. Thus, the bioavailability of T-vc-MMAE after SC administration is at least 50% in mice.

#### 3.3.2. Tumor PK

After IV and SC administration of T-vc-MMAE, $T_{max}$ in the tumor occurred at 72 h for most ADC analytes. Tumor $T_{max}$ for conjugated MMAE after IV injection was around 24~72 h. All ADC analytes showed prolonged exposure in the tumor, and the concentrations were sustained after the peak concentrations. After SC administration of T-vc-MMAE, both $C_{max}$ and ${AUC}_{0-t}$ in the tumors were about 50% (44~67%) of the values observed following IV injection.

After IT administration, the peak concentrations for all the analytes in tumors were observed at the first time point itself. Since unconjugated MMAE concentrations were also high at the first time point, it suggests instant and rapid deconjugation of T-vc-MMAE in the tumor. Considerably higher tumor concentrations of ADC analytes after IT than IV injection were observed for the first three days after dosing, while tumor concentrations became similar between IT and IV routes after three days (Figure 4). Nevertheless, similar to IV and SC routes, prolonged tumor exposures of all ADC analytes were observed for the IT route as well. IT injection also led to a significant increase in tumor $C_{max}$ values for the analytes. $C_{max}$ following IT injection were 96-, 50-, 20-, and 3.5-fold higher than IV injection for conjugated MMAE, total MMAE, total mAb, and unconjugated MMAE, respectively (Table 2). On the other hand, tumor ${AUC}_{0-t}$ values for different ADC analytes were 2~8 times higher for the IT route compared to the IV route (Table 2).

After IV, IT, or SC injection of T-vc-MMAE, the AUC of unconjugated MMAE in tumors was significantly higher than the AUC in plasma (Table 2), validating that ADC can specifically deliver and release the cytotoxic drug at the site-of-action, regardless of the routes of administration. Of importance, IT administration significantly increased tumor-to-plasma exposure ratios for total mAb, total MMAE, and conjugated MMAE compared to IV and SC administration.

Given that exposure at the site-of-action is related to efficacy, IT injection enables significant increases in $C_{max}$ and AUC of multiple ADC analytes in tumors, and thus may enhance drug efficacy compared to the IV route. In contrast, SC administration resulted in around 50% lower $C_{max}$ and AUC values in the tumors, suggesting doubling the dose of ADC may be required to achieve similar efficacy compared to the IV route.

### 3.4. In Vivo Efficacy of T-vc-MMAE Administered via IT, SC, and IV Routes

The PK study results were used to inform the dose selection for the efficacy study, where we aim to match systemic exposure of T-vc-MMAE among IT, SC, and IV routes. With the ratios of plasma ${AUC}_{0-t}$ between IT and IV administration approaching around 100%, the dose levels for IT administration would be identical to the IV administration group. Based on the ${AUC}_{0-t}$ ratio of ~50% between SC and IV routes, doses for SC route would be two times higher than the IV administration group. For each route, three dose levels (high, mid, and low) were included to investigate the efficacy of T-vc-MMAE. For IT and IV groups, the high-, mid-, and low-dose groups would be 10, 3, and 1 mg/kg single dose; for SC group, the doses would be 20, 6, and 2 mg/kg single dose.

The initial tumor volumes were comparable between each route and dose level (Appendix A). Tumor growth curves after T-vc-MMAE administration via IT, SC, and IV routes, along with the control group, are shown in Figure 5. Tumor growth curves stratified by high-, mid-, and low-dose groups are shown in Appendix A. Treatment with T-vc-MMAE at high doses via all three routes resulted in complete tumor regression. Tumor regrowth was observed in mice treated at mid-dose for all three routes, where IT injection delayed tumor regrowth (~45 days after treatment) more efficiently than IV and SC injection (~30 days after treatment). Under mid- and low-dose treatment, IT administration was found to be the most efficacious, whereas SC administration tended to have the least efficacy at low-dose treatment (Appendix A). For the mid-dose group, although SC administration with twice the dose tended to have non-inferior or slightly better efficacy compared to IV administration, it could yield less efficacy compared to the IV route when given at the same dose level.

Higher variability in tumor response to ADC treatment was observed in mice treated with subtherapeutic doses (i.e., mid- or low-dose) intratumorally or subcutaneously (Figure 5). Interestingly, as shown in the individual tumor growth curves, we found that mice with higher or lower tumor volumes treated with high dose T-vc-MMAE IT showed complete response. This may imply that the efficacy of IT route is not affected by initial tumor volume (Figure 5), which is often the case for the IT route. Mice body weights were comparable between all treatment groups and the control group throughout the study period, indicating that the tested dose ranges of T-vc-MMAE administered via IV, IT, or SC routes did not cause systemic toxicity (Appendix A).

### 3.5. Local Toxicity Study of T-vc-MMAE after SC Administration

After SC injection with T-vc-MMAE or naked MMAE, the development of skin lesions in terms of severity and incidence was considerably higher in immunocompetent WT mice than in nude mice (Appendix A). Specifically, after SC treatment with 30 mg/kg single-dose, three of three WT mice developed gross necrosis within 96 h, whereas one of three nude mice presented a relatively mild and smaller skin lesion. Three of three WT mice receiving naked MMAE SC at 0.5 mg/kg single-dose developed gross necrosis that was qualitatively similar but quantitatively less severe than those in WT mice treated with T-vc-MMAE. For nude mice treated with naked MMAE at 0.5 mg/kg single-dose, two of three developed moderate lesions slightly more severe than the observation in nude mice receiving T-vc-MMAE. The development of pathology at the injection site was apparently associated with SC treatment of T-vc-MMAE or MMAE compared with the naked mAb or vehicle control group, where neither of these groups developed pathology for either mouse strain.

The histopathology of selected skin slide sections from WT mice given T-vc-MMAE (Figure 6a), naked MMAE (Figure 6b), naked mAb (Figure 6c), and the control (Figure 6d) is presented in Figure 6. After T-vc-MMAE treatment, skin sections from three of three mice showed hyperparakeratosis, acanthosis, subepidermal blister, and ulceration with full-thickness severe necrosis (involving epidermis and dermis), and moderate infiltration of mixed inflammatory cells in reticular dermis and hypodermis. Mice treated with naked MMAE objectively showed less severe pathology than T-vc-MMAE treatment. Skin sections from three of three mice treated with naked MMAE showed hyperparakeratosis, mild acanthosis, subepidermal blisters, and mild infiltration with mixed inflammatory cells, and only one of three mice developed a small area of ulceration and mild superficial necrosis. No significant pathologic changes were observed in WT mice treated with naked mAb or control.

Histopathology in the skin of nude mice presented less severity than WT mice. After SC administration of T-vc-MMAE, two of three nude mice showed minimal pathological changes, and one of three showed mild acanthosis, mild superficial necrosis, and moderate immune infiltration. One of three nude mice that received naked MMAE SC showed normal skin tissue, while two of three showed parakeratosis, parakeratosis, mild acanthosis, mild superficial necrosis, and moderate immune infiltration.

Semiquantitative analysis results are provided in Figure 6e. WT mice given T-vc-MMAE or naked MMAE developed significant injection site pathology compared to control mice, with tissue damage scores of 9.7 (*p* < 0.0001) and 4.3 (*p* = 0.0005), respectively. In contrast, nude mice receiving T-vc-MMAE or naked MMAE showed no significant difference in tissue damage scores compared to control mice. Thus, SC administration of T-vc-MMAE caused significantly higher tissue damage in WT mice (*p* = 0.005) compared to nude mice, suggesting the role of immune cells in the local toxicity of SC-administered ADCs. In contrast, SC administration of naked MMAE resulted in similar tissue damage between two mouse strains (*p* = 0.624). Interestingly, WT mice given ADC showed significantly higher local toxicity than those given molar equivalent MMAE (*p* = 0.02), suggesting enhanced local toxicity of ADC due to prolonged local exposure or enhanced immune cell uptake of payload conjugated to the mAb.

### 3.6. Development of a PK/PD Model for T-vc-MMAE Administered via IT, SC, and IV Routes

Figure 7 shows the observed data and model-fitted PK profiles of total mAb, total MMAE, and unconjugated MMAE in plasma and tumor, after IT, SC, and IV administration of T-vc-MMAE. The model is able to simultaneously capture the PK profiles of different ADC analytes in plasma and tumors for all three routes of administration. As shown in Table 1, all PK parameters are estimated with good precision (%CV < 30) at each step.

The volume of distribution of mAb in the central compartment (${V1}_{mAb}$) estimated in step 1 corresponds to the reported mouse plasma volume, confirming the distribution of the mAb component of an ADC is primarily restricted in the systemic circulation. In contrast, the estimated volume of distribution of unconjugated MMAE in peripheral tissue (${V2}_{PL}$) is relatively large, suggesting an extensive tissue distribution of free MMAE. Importantly, the estimated $f_{dec}$ is 30.8 (%CV 20.7), implying the deconjugation rate of T-vc-MMAE in the tumor microenvironment can be ~30-fold higher than the rate in plasma.

Regarding route-specific parameters, the model estimated $F_{IT}$ value was 0.75 (%CV 4.61), which indicates the tumors can hold ~75% of the injection IT dose, while ~25% of the injected dose immediately distributes into the systemic circulation. Since the total IT injected volume is 14 μL for each mouse, the NCI-N87 tumors may accommodate 6~9 μL of injected volume. The absorption rate constant ($k_{a}$) and SC bioavailability (F) of T-vc-MMAE were estimated with good precision (%CV < 2). The model estimated F value (~50%) corresponds to the NCA result calculated using observed PK data (Table 2).

Figure 8 shows the model-predicted individual tumor growth curves superimposed over the observed PD data for IT, IV, and SC administration routes. The model was able to capture all PD data well, upon accounting for the variability in linear tumor growth rate, maximum killing rate, and cell death kinetics among individual mice. The PD parameters were estimated with good precision (i.e., %CV < 25 for all parameters), as shown in Table 1. The estimated ${\mathrm{EC}}_{50}$ value for %TE was 17.9% (%CV 1.17), indicating when ~20% of tubulin is occupied by MMAE, it can exert 50% of the maximum tumor killing effect.

### 3.7. PK/PD Model Simulations for Administration Route-Dependent Dose Optimization

Simulations of TGI curves using clinically approved dosing regimens for MMAE-based ADCs (1.8 mg/kg Q3W) show that treatment via the IV route results in tumor stasis, and the IT route enables complete tumor remission. In contrast, the SC route fails to inhibit tumor growth (Figure 9a). The simulation results also show that IT administration of T-vc-MMAE at the dosing regimen of 0.3 mg/kg Q3W (Figure 9b) or 1.8 mg/kg Q8W (Figure 9c) can achieve similar antitumor activity to the currently approved IV dosing regimen. Thus, IT administration may allow decreasing the dose by 6-fold or extending dose frequency from Q3W to Q8W while maintaining similar efficacy as IV administration, and likely decreasing the toxicity. In contrast, the simulation results indicate that treatment of T-vc-MMAE subcutaneously requires a higher dose amount (4.0 mg/kg Q3W, Figure 9b) or more frequent dosing (1.8 mg/kg QW, Figure 9c) to achieve similar efficacy as an IV injection. Notably, during this simulation exercise, we found that even a slight change in IT dose level could significantly affect ADC’s antitumor activity, which emphasizes the steep dose–response relationship for ADCs administered via the IT route.

## 4. Discussion

The modulation of the administration route has the potential to enhance the therapeutic index of biotherapeutics [8,24,50,51]. However, for ADCs, apart from the conventional IV route, limited studies have explored the therapeutic potential of alternative routes. Consequently, here we conducted preclinical PK/PD studies to examine the feasibility of SC and IT routes for ADCs, using T-vc-MMAE as the model compound. PK profiles of ADC in plasma and tumors, together with efficacy data after IV, IT, and SC administration, were generated to facilitate the establishment of a robust exposure–response relationship for the MMAE-based ADC. A semi-mechanistic PK/PD model was developed to simultaneously characterize PK and TGI data for IV-, IT-, and SC-administered T-vc-MMAE. The model was also used to investigate and optimize clinically approved IV dosing regimens for IT and SC routes. Additionally, we objectively evaluated the local toxicity of the ADC administered via the SC route in immunocompetent and immunodeficient mice. As such, the preclinical investigation presented here provides unprecedented insight into the PK/PD behavior of ADCs administered via IT and SC routes, and paves the way for the evaluation of these routes in the clinic.

Since we generated plasma and tumor PK data for multiple ADC analytes after IV, IT, and SC administration, it was possible to assess how the route of administration affects systemic and site-of-action PK of ADCs. IT administration led to the significantly higher tumor exposure of all ADC analytes, and relatively higher concentrations were observed for the first three days after dosing. In plasma, the IT route resulted in lower $C_{max}$ and similar AUC of ADC analytes compared to the IV route. After SC administration, delayed and significant decreases of peak concentrations of ADC analytes in plasma were observed, and plasma and tumor AUC of different ADC analytes were around 50~80% of the values after IV administration.

Identifying which exposure matrices (i.e., $C_{max}$, AUC, or $C_{trough}$) drive efficacy and toxicity is essential to establish E-R relationships, and to investigate how different administration routes may affect the therapeutic index for ADCs. E-R analysis based on clinical data of various MMAE-based ADCs, where only plasma but not site-of-action PK data are available, suggests that plasma AUC of conjugated MMAE (acMMAE) correlates to efficacy and probability of peripheral neuropathy; whereas plasma $C_{trough}$ correlates to peripheral neuropathy but not efficacy [52,53,54,55,56]. Therefore, the observed decreased $C_{max}$ of conjugated MMAE in plasma after SC and IT injection compared to IV administration may not be able to reduce the toxicity of peripheral neuropathy. In addition, the intention of utilizing SC administration to facilitate more frequent doses and benefit from fractionated dosing regimens may not be beneficial for $C_{trough}$ driven toxicity, since frequent dosing results in higher $C_{trough}$ and possibly increased toxicity [57]. We found IT administration of T-vc-MMAE had better efficacy than IV and SC routes, which could be explained by different tumor PK. The enhancement of tumor $C_{max}$ or tumor to plasma AUC ratio after IT injection was observed to be the highest for conjugated MMAE, which suggests achieving high concentrations of acMMAE in tumors is more important for ADC’s efficacy.

The semi-mechanistic PK/PD model developed here well characterized the PK profiles of different ADC analytes in plasma and tumors, and also TGI data after IV, IT, and SC administration of T-vc-MMAE in mice. While we did not perform the PK of naked mAb and MMAE in this study, previously reported plasma PK data of trastuzumab [32] and free MMAE [47] were used to develop the PK models for mAb and payload components of ADC. PK data for different analytes of T-vc-MMAE obtained after administration of ADC via different routes were used to refine route-specific parameters. The model-estimated SC bioavailability for T-vc-MMAE was ~50%, supporting the dose selection for the efficacy study. The model estimated that NCI-N87 tumors (~200 mm^3^) could hold ~75% (about 6~9 μL) of injected IT dose, which corresponds well with the reported hold-up volume of B16F10 tumors (about 6.6~13.3 μL) [58].

The present mechanistic tumor disposition model also has unique features that distinguish it from other existing models. It allowed us to characterize concentrations of ADC analytes at the site-of-action and obtain %TE. The %TE, rather than plasma concentration or dose, was then used to drive the efficacy of ADC. In addition, the interaction between PK and PD models is dynamic, where the real-time tumor size determines the MMAE tumor concentration and %TE, which in turn affect the tumor size. The model suggested that ~20% of β-tubulin bound with payload could induce ~50% of the maximum killing effect. Since the transit compartment model could characterize the delayed ADC effect, the model suggests that ADC efficacy may be determined mainly by achieving sufficient %TE (i.e., concentration-dependent) rather than duration of tumor exposure (time-dependent). This also implies that $C_{max}$ in tumors can be a therapeutically relevant PK parameter.

The current PK/PD model for T-vc-MMAE could facilitate preclinical to clinical translation and can be used for clinical trial simulations following the adjustment of PK parameters and system-specific PD parameters to clinically plausible values, while keeping PK/PD model structure the same [38,59]. PK parameters can be calibrated by fitting the PK model to the clinically observed data or allometric scaling from the preclinical species, and PD parameters (i.e., tumor growth rate, initial and maximum tumor burdens) can be adjusted to clinically plausible values. The translated PK/PD model can be used to predict clinical outcomes such as efficacy endpoints [38], clinically efficacious doses [60], and the observed antitumor objective responses (i.e., complete response, disease-free survival, overall survival) [61], which can be compared with the reported clinical trial data to validate the performance of the PK/PD model. More specifically, the present PK/PD model can be optimized using clinical PK/PD data from MMAE-based ADCs administered via the IV route, and then be utilized to investigate and compare clinical efficacy for different administration routes with different dosing regimens.

In addition to the efficacy, the local toxicity of SC-administered T-vc-MMAE, along with the toxicity of SC mAb and MMAE, was examined at 96 h after dosing in both immunocompetent and immunocompromised mice. T-vc-MMAE dose used for the local toxicity study was based on the lower bound of the maximum tolerated dose reported for MMAE-based ADCs (30~40 mg/kg) [62]. The dose of naked MMAE was equivalent to the molar dose of T-vc-MMAE, when the average DAR of ~3.4 and SC bioavailability for MMAE of ~86% is considered [63]. The local toxicity study for ADC presented here is unprecedented and has some key features. First, we included ADC and its naked mAb and payload groups for comparison, which allowed us to identify which ADC component (i.e., mAb or payload) contributes to ADC’s local toxicity. The significant tissue damage observed in MMAE and T-vc-MMAE groups, and the intact tissue observed in the mAb group, indicates that the payload component is primarily responsible for the local toxicity of T-vc-MMAE. Second, the utilization of immunocompetent and immunodeficient mice can provide insight into the mechanisms of local toxicity for T-vc-MMAE. We found significantly more severe tissue damage in WT mice than in nude mice, suggesting the involvement of skin immune responses [64]. This observation also highlights the importance of in vivo model selection to examine local tolerance of ADCs, as, in our study, nude mice failed to detect substantial local toxicity of SC ADC. Thus, a previous study based on nude mice data stating that SC ADC was well-tolerated may require further investigation in an immunocompetent in vivo system [18]. Third, our study highlights that PK properties of molecules (i.e., absorption) may also affect the severity and incidence of local toxicity. We surprisingly found that SC administration of T-vc-MMAE results in significantly greater local toxicity than SC administration of MMAE. Since smaller drugs (<1 kDa) mostly undergo fast blood capillary absorption and larger therapeutic proteins (>16 kDa) predominantly undergo slow lymphatic absorption [65], the slower absorption process and longer local retention of large molecule ADC can cause higher toxicity than small molecule MMAE [64]. Moreover, significantly different toxicity between WT and nude mice was found after ADC treatment but not MMAE treatment, indicating prolonged local drug exposure may induce immune system and skin inflammation [66]. Both qualitative and quantitative analyses were applied to analyze the results from the local tolerance study. Initial observation of skin at the injection site, followed by microscopic and semiquantitative analyses by pathologists blinded to the treatment, enabled objective evaluation of ADC toxicity. the observed data suggest that SC route may lead to local skin toxicity, and one needs to be careful while switching the route of administration from IV to SC for MMAE-based ADCs.

Our data suggest that IT route may be superior to conventional IV route for ADCs. This is based on the presented quantitative PK/PD analysis, as well as current advancement of IT injection technique and emerging clinical evidence of IT immunotherapy benefits [67,68,69]. Our PK/PD data indicated that local delivery of ADCs may enhance antitumor activity (Figure 5). PK/PD model simulations further suggest that IT route allows dose reduction by 6-fold or dose frequency extension from Q3W to Q8W compared to IV route for MMAE-based ADCs (Figure 9).

While IT injection of drugs is not novel, it has rarely been favored by clinicians, because the IT injection technique is challenging, and it restricts the treatment to certain tumor types with palpable cutaneous lesions (i.e., melanoma) [68]. Nonetheless, the utilization of image-guided injection enables more accurate delivery and expansion of the treatment to deeper tumors. Indeed, recent clinical studies have investigated ADC and ICI combination therapy delivered IT [70,71]. There is emerging clinical evidence that suggests IT immunotherapies may provide advantages such as reduced drug amounts and off-target toxicity, stronger antitumor activity in the injected tumors and distant noninjected sites, and overcoming ICI resistance [24,69]. As such, IT delivery of ADC and ICI combination therapy may be a potential novel strategy for treating cancer.

Nonetheless, several issues specific to IT delivery need to be considered. First, repeated and frequent needle punctures during IT injection may cause risks of bleeding and injury [72]. Second, with higher vascularity in some visceral organs (i.e., liver, lung, heart) or tumors [73], IT delivery may have the risk of intravasation of the injected drug and cause systemic toxicity. Moreover, the variability during IT injection procedure (i.e., different physicians, needle design, imperfect injection) may cause substantial differences in the treatment outcome [72]. However, standardized methods for IT delivery have not yet been established [22]. Importantly, the variability of the IT injection procedure may yield variable dosing amounts and consequently lead to inconsistent outcomes [74], which can be exacerbated for drugs with a narrow therapeutic index such as ADCs. This tendency is also observed during our PK/PD model simulations, where a slight change in ADC dose significantly affected TGI curves. Lastly, the imperfect IT injection may lead to reduced drug delivery into the tumors and immediate drug distribution into the systemic circulation, resulting in reduced therapeutic window for ADCs following IT administration [74].

There are also some limitations of our study that need to be considered. First, the dose ranges of T-vc-MMAE ADC used in the efficacy study were unable to induce systemic toxicity for comparison between different routes of administration. Since trastuzumab does not bind to mouse HER2, additional studies in human HER2-expressing mice may be needed to evaluate on-target off-tumor systemic toxicity of T-vc-MMAE ADC. In addition, to truly demonstrate IT delivery can broaden the therapeutic index of ADCs, which is an increase in efficacy (as shown in this study) while maintaining (or mitigating) safety risks, further studies on systemic toxicity (e.g., off-target toxicity, on-target off-tumor toxicity) in mice or higher species after IT administration would be needed. Second, the PK of each ADC analyte at the SC injection site was not measured. The observation that ADC was more locally toxic than its free payload suggests local PK of ADC affects the severity of local intolerance. It has been reported that >30 protease enzymes are present in the skin, where cysteine proteases (i.e., cathepsin-B) form the major fraction of the total proteolytic enzymes [21]. These proteolytic enzymes can cause presystemic catabolism or deconjugation, resulting in payload cleavage and local tissue damage [19,20,21]. Thus, the presence of proteases locally can lead to significantly greater SC toxicity of ADCs, as seen in our study. Therefore, to better understand the reasons for local toxicity of ADCs, and to devise strategies to overcome this toxicity (e.g., protease inhibitors in the formulation) [75], quantitative analysis of ADC PK at the SC injection site is warranted.

The relevance and translatability of our findings to the clinic remain to be seen. The scalability of findings from preclinical species to the clinic is often challenging for local delivery routes [13]. For example, SC bioavailabilities of mAb are known to have weak correlations between animals and humans [19]. Tumor microenvironment can differ markedly across species, and thereby local and systemic drug exposure after IT delivery would alter accordingly [24]. Therefore, the enhancement of efficacy for IT-delivered ADC observed in our mouse xenograft requires further validation in tumors with a clinically-relevant size and relevant tumor heterogenicity. In addition, the severity of local toxicity for SC-delivered MMAE-based ADC may need confirmation in higher species. Notably, the current inferences made for MMAE-based ADC may also change for different conjugation methods and types of linker-payload. Nonetheless, the findings presented here provide an important cornerstone for the clinical evaluation of ADCs following SC and IT administration.

## 5. Conclusions

Here we found that IT administration of ADC significantly increased tumor ADC exposure and enhanced anti-tumor activity in vivo. Additionally, the model simulation suggests that switching from IV to IT injection allows for dose frequency extension from Q3W to Q8W or dose level reduction by ~6-fold. Thus, IT injection can potentially improve the therapeutic index of ADCs compared to conventional IV injection. On the other hand, in vivo efficacy of SC treatment was inferior to IV treatment. In addition, local toxicity was observed after SC administration of ADC. These data suggest that SC delivery of MMAE-based ADCs to treat cancer may be challenging. Thus, this manuscript presents a critical evaluation of novel administration routes for ADCs using preclinical models. The findings from this manuscript are expected to facilitate further clinical evaluation of ADCs administered via SC and IT routes.

## Figures and Tables

**Figure 1 pharmaceutics-15-01132-f001:**
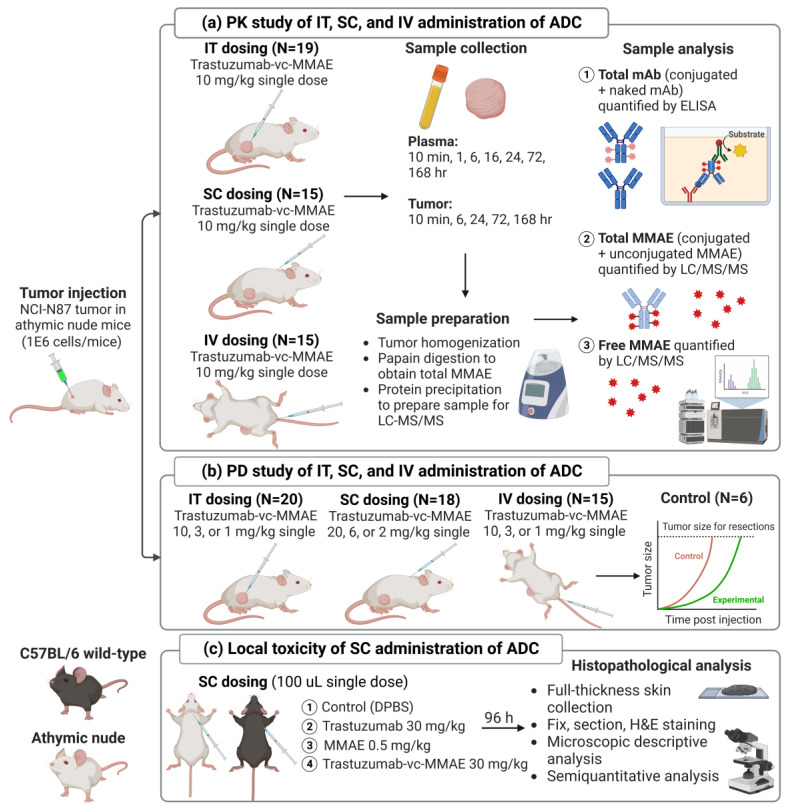
An overview of the experimental design. (**a**) PK study following IT, SC, and IV administration of 10 mg/kg T-vc-MMAE in NCI-N87 tumor-bearing nude mice; (**b**) efficacy study of IT, SC, and IV administered T-vc-MMAE (high, mid, and low doses) in NCI-N87 tumor-bearing nude mice; (**c**) local toxicity study of SC administered T-vc-MMAE, naked mAb (trastuzumab), naked payload (MMAE), and vehicle control in wild-type and nude mice.

**Figure 2 pharmaceutics-15-01132-f002:**
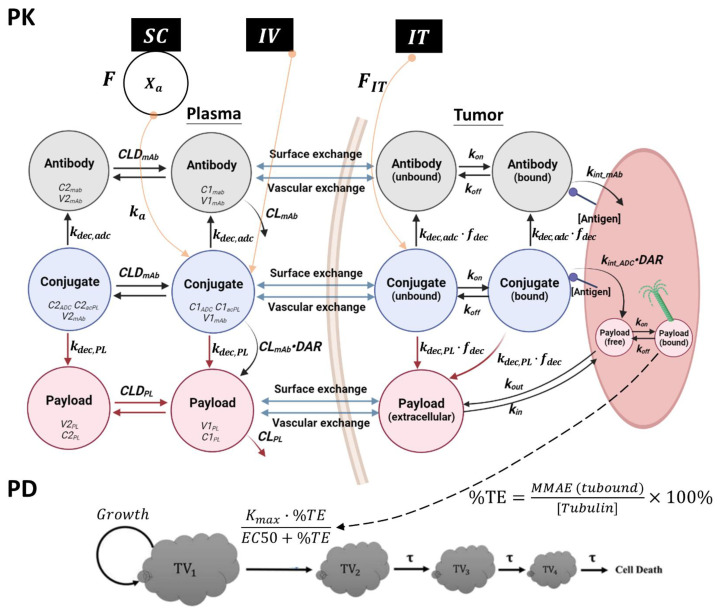
Schematic diagram of the semi-mechanistic PK/PD model developed for IV, IT, and SC administered T-vc-MMAE ADC. The PK model consists of the plasma and tumor PK models connected via vascular (extravasation) and surface exchange (diffusion). The PK model is connected to the PD model using intracellular target engagement (%TE) predicted by the PK model, which is used to drive the efficacy of T-vc-MMAE. Please refer to the PK and PD model structure sections in the method section for a detailed description of the symbols and disposition processes captured by the model.

**Figure 3 pharmaceutics-15-01132-f003:**
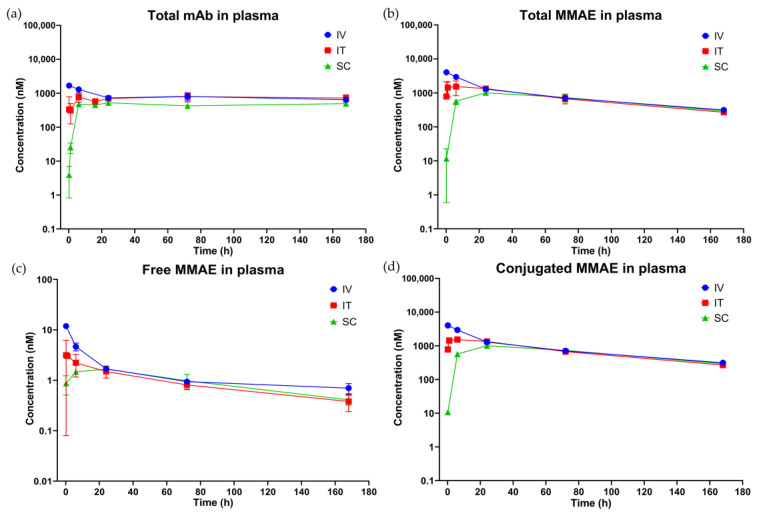
Observed plasma pharmacokinetics (PK) of ADC analytes in mice after intravenous (IV), intratumoral (IT), and subcutaneous (SC) administration of 10 mg/kg of T-vc-MMAE single dose. The figure displays the mean (SD) observed concentrations of: (**a**) total antibody; (**b**) total MMAE; (**c**) unconjugated MMAE; and (**d**) conjugated MMAE in plasma.

**Figure 4 pharmaceutics-15-01132-f004:**
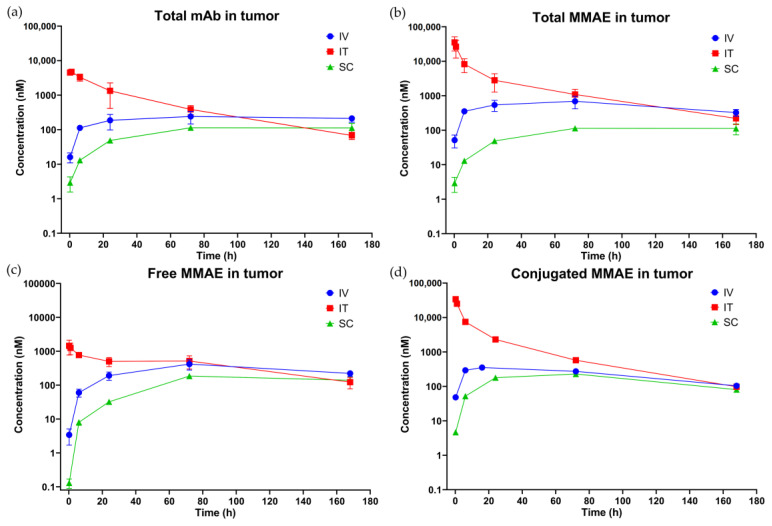
Observed tumor pharmacokinetics (PK) of ADC analytes in mice after intravenous (IV), intratumoral (IT), and subcutaneous (SC) administration of 10 mg/kg of T-vc-MMAE single dose. The figure displays the mean (SD) observed concentration of: (**a**) total antibody; (**b**) total MMAE; (**c**) unconjugated MMAE; and (**d**) conjugated MMAE in the tumor.

**Figure 5 pharmaceutics-15-01132-f005:**
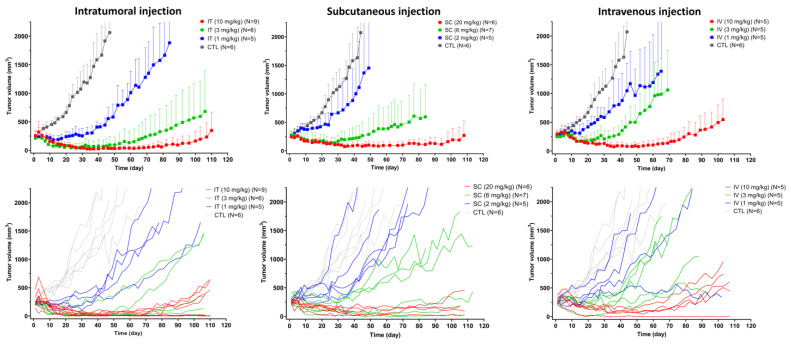
In vivo efficacy of T-vc-MMAE ADC after intravenous (IV), intratumoral (IT), and subcutaneous (SC) administration. The figures show the mean (SD) tumor growth curves (upper) and individual tumor growth curves from each animal (lower) after IT (10, 3, 1 mg/kg), SC (20, 6, 2 mg/kg), and IV (10, 3, 1 mg/kg) administration of T-vc-MMAE single dose, along with the untreated group.

**Figure 6 pharmaceutics-15-01132-f006:**
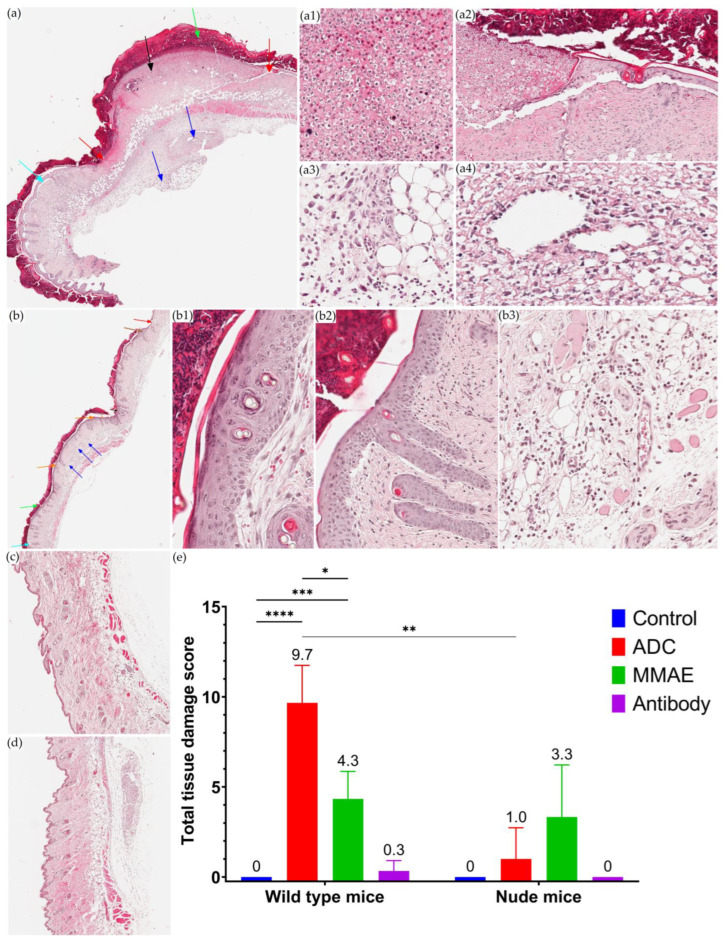
Local toxicity of subcutaneously administered ADC in wild-type mice. Representative histopathology in wild-type mice that received: (**a**) 30 mg/kg of T-vc-MMAE single-dose subcutaneously, (**b**) 0.5 mg/kg of MMAE single dose subcutaneously, (**c**) 30 mg/kg of trastuzumab single dose subcutaneously, and (**d**) vehicle control subcutaneously. The microscope magnification was 20×. Tissue damage is indicated by arrows displaying necrosis (black), ulceration (red), inflammatory infiltrate (blue), hyperkeratosis (green), blisters (cyan), acanthosis (brown), and hypergranulosis (orange). The tissue damage shown includes necrosis (**a1**), ulceration (**a2**), inflammatory infiltrate (**a3**,**a4**) are shown. The represented tissue damage of hypergranulosis (**b1**), acanthosis (**b2**), inflammatory infiltrate (**b3**) are shown. (**e**) The histogram displays total tissue damage scores (mean and SD) calculated by summation of severity scores (0~3) of blister formation, inflammatory infiltrate, ulceration, and necrosis; * *p* < 0.05, ** *p* = 0.005, *** *p* = 0.0005, **** *p* < 0.0001.

**Figure 7 pharmaceutics-15-01132-f007:**
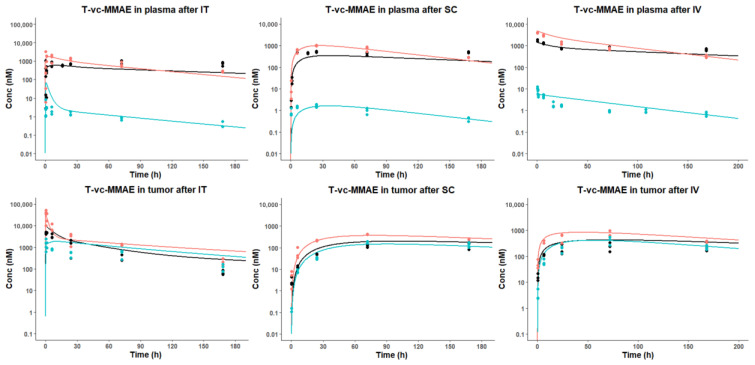
Comparison of model fitted and observed PK profiles of T-vc-MMAE analytes in plasma and tumors after intravenous (IV), intratumoral (IT), and subcutaneous (SC) administration of 10 mg/kg of T-vc-MMAE single dose. The figure displays observed (dots) and model-predicted (solid lines) plasma and tumor concentration vs. time profiles of total antibody (red), total MMAE (black), and unconjugated MMAE (cyan) in mice.

**Figure 8 pharmaceutics-15-01132-f008:**
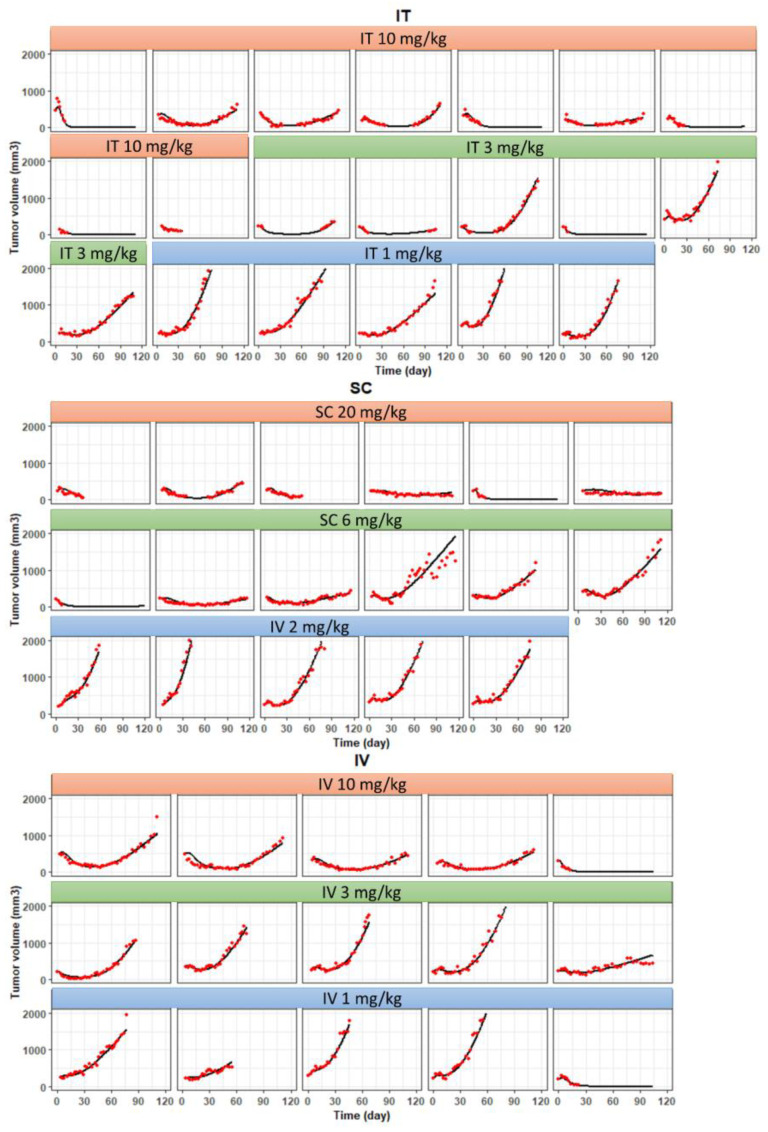
Comparison of PK/PD model fitted and observed tumor growth curves after treatment with T-vc-MMAE administered via intravenous (IV), intratumoral (IT), and subcutaneous (SC) routes. The figure displays observed (red dots) and model-predicted (solid lines) tumor growth curves after treatment with high (red bar), mid (green bar), and low (blue bar) ADC doses.

**Figure 9 pharmaceutics-15-01132-f009:**
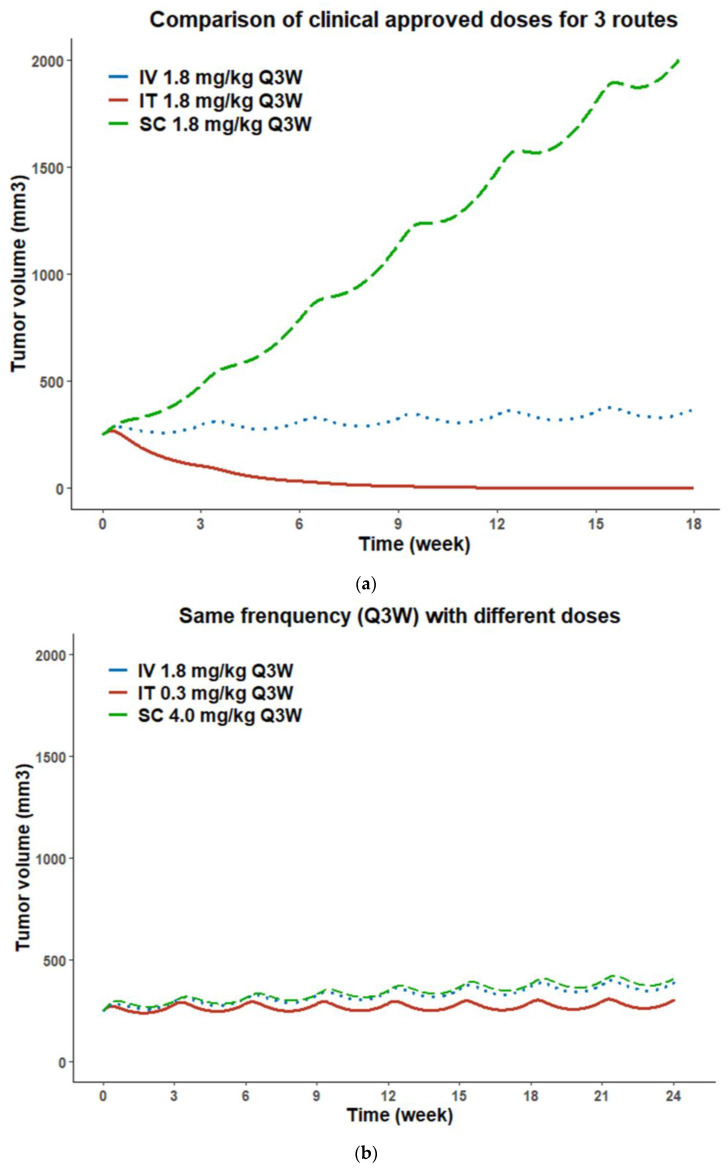
PK/PD model simulated tumor growth inhibition (TGI) after treatment with T-vc-MMAE administered via intravenous (IV), intratumoral (IT), or subcutaneous (SC) routes. (**a**) Simulation for clinically approved dosing regimens of 1.8 mg/kg given Q3W for 6 cycles; (**b**) simulation of dosing amounts required to achieve similar TGI after IV, IT, and SC administration with the same dosing frequency (Q3W); (**c**) simulation of dosing frequency to achieve similar TGI after IV, IT, and SC administration with the same dosing amount (1.8 mg/kg).

**Table 2 pharmaceutics-15-01132-t002:** Plasma and tumors PK parameters for ADC analytes observed after intravenous (IV), intratumoral (IT), and subcutaneous (SC) administration of 10 mg/kg of T-vc-MMAE ADC single dose.

	$T_{max}$ (h)	$C_{max}$ (nM)	${AUC}_{0-t}$ (h·nM)	${AUC}_{inf}$ (h·nM)	IT/IV and SC/IV $C_{max}$ Ratio (%)	IT/IV and SC/IV ${AUC}_{0-t}$ Ratio (%)	Tumor to Plasma ${AUC}_{0-t}$ Ratio (%)
Total antibody (measured as conjugated + unconjugated antibody)							
Plasma							
IV	0.167	1690 (5.87)	1.34 × 10^5^ (4.12)	7.48 × 10^5^	-	-	-
IT	72	798 (17.6)	1.24 × 10^5^ (9.31)	8.49 × 10^5^	47.2	92.5	-
SC	24	524 (3.73)	7.70 × 10^4^ (3.98)	- ^2^	31.0	57.4	-
Tumor							
IV	72	242 (22.7)	3.52 × 10^4^ (12.8)	1.94 × 10^5^	-	-	26.3
IT	1	4739 (3.16)	1.30 × 10^5^ (14.6)	1.33 × 10^5^	1959	369	104
SC	72	114 (7.88)	1.54 × 10^4^ (8.24)	- ^2^	47.0	43.7	20.0
Total MMAE ^1^ (measured as conjugated + unconjugated MMAE)							
Plasma							
IV	0.167	4058 (3.37)	1.28 × 10^5^ (3.39)	1.52 × 10^5^	-	-	-
IT	6	1538 (26.5)	1.28 × 10^5^ (7.97)	1.52 × 10^5^	37.9	99.8	-
SC	24	1011 (2.12)	1.07 × 10^5^ (7.51)	1.41 × 10^5^	24.9	83.4	-
Tumor							
IV	72	693 (22.9)	8.79 × 10^4^ (13.9)	1.74 × 10^5^	-	-	68.5
IT	0.167	3.54 × 10^4^ (19.7)	3.75 × 10^5^ (13.6)	3.87 × 10^5^	5111	426	293
SC	72	415 (1.24)	4.82 × 10^4^ (2.20)	8.17 × 10^4^	59.8	54.8	45.0
Unconjugated MMAE							
Plasma							
IV	0.167	11.9 (5.04)	224 (3.50)	324	-	-	-
IT	0.167	3.15 (43.6)	161 (7.64)	200	26.4	72.1	-
SC	24	1.67 (8.93)	165 (8.89)	207	14.0	73.9	-
Tumor							
IV	72	419 (19.3)	4.79 × 10^4^ (12.6)	8.16 × 10^4^	-	-	2.14 × 10^4^
IT	0.167	1449 (20.9)	7.30 × 10^4^ (13.2)	8.59 × 10^4^	346	207	4.52 × 10^4^
SC	72	185 (6.13)	2.12 × 10^4^ (6.28)	6.87 × 10^4^	44.2	60.3	1.28 × 10^4^
Conjugated MMAE							
Plasma							
IV	0.167	4046 (3.37)	1.28 × 10^5^ (3.40)	1.51 × 10^5^	-	-	-
IT	6	1536 (26.5)	1.28 × 10^5^ (7.98)	1.52 × 10^5^	38.0	99.8	-
SC	24	1009 (2.13)	1.07 × 10^5^ (7.51)	1.41 × 10^5^	25.0	83.4	-
Tumor							
IV	24	353 (23.0)	4.01 × 10^4^ (16.9)	5.25 × 10^4^	-	-	31.3
IT	0.167	3.40 × 10^4^ (19.8)	3.02 × 10^5^ (15.4)	3.06 × 10^5^	9635	753	236
SC	72	230 (5.75)	2.70 × 10^4^ (3.98)	3.44 × 10^4^	65.1	67.3	25.2

^1^ Measured as the sum of conjugated and unconjugated MMAE concentrations. ^2^ The % extrapolated of AUC is >20% and hence AUC_inf_ is not reported.

## Data Availability

Data generated are included within the manuscript.

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
