# Peer review of "Pharmacokinetics and Pharmacodynamics of Antibody-Drug Conjugates Administered via Subcutaneous and Intratumoral Routes"

_pharmaceutics, 2023, doi:10.3390/pharmaceutics15041132_

Round 1

Reviewer 1 Report

This paper provides comprehensible information about the use of pharmacokinetic and pharmacodynamic models of ADSCs for obtaining a better therapeutic index when used in anticancer therapies. The paper is very well written, presenting innovative research in cancer, by applying mathematical modeling.

However, my main concerns are the missing PKPD model from the paper and the applicability of it by the clinicians given the complexity and uncertainty of parameters. Please refer to the following comments while revising the manuscript:

1. Please define all the abbreviations when first mentioned in the text (e.g., ADC..).

2. Please provide a reference for the HIC method line 154. 

3. Do you have an explanation for the sacrification of mice at the time specified in lines 176-177? Why did you choose this time? Why was the sacrifice needed?

4. You mentioned that "Tumor volume and body weight were monitored 3 times per week for ~110 days or until tumor volume reached > 2000 mm3." However, later you write that " After 96 h of SC dosing, all animals were euthanized,.." From this, I understand that while measuring the tumor volume for maximum 110 days, only 4 days were actually necessary since this is the only time when ADC was administrated. How did the tumor volume was influenced during these 4 days?

5. The paper title is "PK/PD..", hence I expect to find a model in the paper text, not in the Appendix. Where is the appendix? I cannot find it in the pdf or supplementary material. Moreover, you can not refer to the number of equations in the text if they cannot be found in the main text. 2.7.1. provides a comprehensive explanation of PK model structure, without having any model as a reference until that point.

6. I found the PK/PD model too difficult to be used in clinical practice, given the number of parameters that increase the uncertainty (more than 40 parameters in Table 1???).

Please add in the Discussion section how you see the application of your research in clinical practice.

How are the clinicians going to implement the model? What further trials are necessary to validate the proposed model?

Also, I find it necessary to compare you PKPD with other PKPD models developed for cancer and applied in clinical practice(A Minimal PKPD Interaction Model for Evaluating Synergy Effects of Combined NSCLC Therapies, doi:10.3390/jcm9061832, Model Calibration of Pharmacokinetic-Pharmacodynamic Lung Tumour Dynamics for Anticancer Therapies, https://doi.org/10.3390/jcm11041006, Lung Tumor Growth Modeling in Patients with NSCLC Undergoing Radiotherapy,https://doi.org/10.1016/j.ifacol.2021.10.261).

7. In Figure 5 the results are shown for 120 days. However, it was mentioned that the therapy was administrated for 4 days, then the mice were sacrificed. Please correct where necessary.

8. Moreover, for the PD model you measure the tumor volume. How do you plan to measure the tumor volume in a human patient? It cannot be possibly performed every 3 days as you mentioned. Most probably you will receive a tumor measurement at 3 months via CT scans. You will only have 2 points (day one and day 120), so how can you validate your model by having only 2 points, which usually form a line, not an exponential model like yours?

9. How can you calibrate and validate every parameter in clinical practice on a patient? Please give several mathematical practices (or clinical) to measure/calculate/calibrate the presented parameters on a mouse to a patient.

Reviewer 2 Report

The presented work describes results from preclinical studies assessing the PK, efficacy and local tolerability of an ADC administered via various routes. The pharmacokinetic profile of trastuzumab-vc-MMAE, a model ADC compound, and its analytes in plasma and tumor, following IV, SC and IT administration were examined. Also, the efficacy of trastuzumab-vc-MMAE following administration by the aforementioned routes was assessed in a mouse xenograft model. Moreover, the local tolerability was evaluated in immunocompetent and immunodeficient mice following SC administration. Additionally, the comprehensive preclinical results were used to develop a semi-mechanistic PK/PD model, which was utilized to investigate clinical dosing regimens of MMAE-based ADCs. 

Overall, I commend the authors efforts in performing a detailed investigation to determine the preclinical PK/PD/local tolerability of a model ADC compound, developing a fit-for-purpose mechanistic model, and putting together a well written manuscript. I have the following comments/suggestions for the authors.

1.     Many clinical antibody-based therapies do not cross-react with mouse targets. Does trastuzumab bind to mouse target?  If no, is mouse even a relevant preclinical tox species for safety evaluation to understand systemic toxicity? Given that TI and safety are discussed at several places in the text, the authors need to provide this information and the relevance of safety testing in mouse model for this model ADC compound. If needed, the authors should revisit and modify the relevant text in the manuscript to appropriately address this.

a.     For instance, the authors describe certain limitations of their work in the discussion section. 

“First, the dose ranges of T-vc-MMAE ADC used in the efficacy study were unable to induce systemic toxicity for comparison between different routes of administration. To truly demonstrate IT delivery can broaden the therapeutic index of ADCs, which is increase efficacy (as shown in this study) while maintaining (or mitigating) safety risks, further studies of systemic toxicity after IT administration at higher doses may be needed.”

While assessing local tolerability in a non-binding mouse model is not inappropriate, the above discussion would only be relevant if trastuzumab binds mouse target. If that is not the case, mouse models would have extremely limited value in providing insights on the safety front. As such, even testing at higher dose levels would be of little value to inform safety.

2.     Suggest including a brief description of T-vc-MMAE in the introduction.

3.     In Fig 1c, semiquantitative is misspelt. 

4.     In Table 2, please check for accuracy of AUCinf values for total antibody in plasma and tumor following SC administration as SC AUCinf seems to be higher than IV AUCinf. This is not usually expected as the dose level is the same for IV and SC, and also the SC bioavailability is reported to be ~50-80%.

5.     Please rephrase the below sentence in results for better clarity. 

“Under mid- and low-dose treatment, IT administration was found to be the most efficacious, whereas SC administration tends to have the least efficacy (Figure S5.” 

While this is true for low dose groups, it doesn’t seem to be true for mid-dose groups at the administered dose levels in the efficacy study. In mid-dose groups (fig 5. - green profiles), it looks like efficacy following SC administration is slightly better than IV dosed. The sentence could be true if SC and IV dose levels are similar. The statement should be rephrased to make this clear.
